# NUWA-Infinity: Autoregressive over Autoregressive Generation for Infinite Visual Synthesis

**Jian Liang**[1*]   **Chenfei Wu**[2*]   **Xiaowei Hu**[3]   **Zhe Gan**[3]   **Jianfeng Wang**[3]
**Lijuan Wang**[3]   **Zicheng Liu**[3]   **Yuejian Fang**[1†]   **Nan Duan**[2†]
[1]Peking University   [2]Microsoft Research Asia   [3]Microsoft Azure AI
{j.liang@stu,fangyj@ss}.pku.edu.cn
{chewu,xiaowei.hu,zhe.gan,jianfw,lijuanw,zliu,nanduan}@microsoft.com

## Abstract

Infinite visual synthesis aims to generate high-resolution images, long-duration videos, and even visual generation of infinite size. Some recent work tried to solve this task by first dividing data into processable patches and then training the models on them without considering the dependencies between patches. However, since they fail to model global dependencies between patches, the quality and consistency of the generation can be limited. To address this issue, we propose NUWA-Infinity, a patch-level *"render-and-optimize"* strategy for infinite visual synthesis. Given a large image or a long video, NUWA-Infinity first splits it into non-overlapping patches and uses the ordered patch chain as a complete training instance, a rendering model autoregressively predicts each patch based on its contexts. Once a patch is predicted, it is optimized immediately and its hidden states are saved as contexts for the next *"render-and-optimize"* process. This brings two advantages: ($i$) The autoregressive rendering process with information transfer between contexts provides an implicit global probabilistic distribution modeling; ($ii$) The timely optimization process alleviates the optimization stress of the model and helps convergence. Based on the above designs, NUWA-Infinity shows a strong synthesis ability on high-resolution images and long-duration videos. The homepage link is `https://nuwa-infinity.microsoft.com`.

## 1 Introduction

The field of deep visual synthesis has witnessed great advances in recent years, with a notable trend of generating images from low to high resolutions. Specifically, many works have shifted from generating images with resolution of $256 \times 256$ [25, 36, 10, 37] to $512 \times 512$ [7] and $1024 \times 1024$ [24], or even infinite ones of $256 \times \infty$ [29] and $512 \times \infty$ [17, 2]. The same trend is witnessed in the video field as well, as the number of supported frames is increased from 25 [35] and 48 [4] to even infinity [18]. By increasing the spatial resolution and the number of temporal frames to infinity, these models have shown great visual quality with both high fidelity and creativity.

To support "infinity", most works follow the *divide and conquer* strategy to first divide a large image into several processable patches and then train them in a separate way. GAN-based models [32, 29] attempt to divide large images into several patches and optimize each of them from global or coordinate latents separately. Since different regions have no explicit dependency, these models struggle to merge different patches during inference and can easily lead to inconsistent results. To address this issue, auto-regressive models [8, 2] incorporate a sliding window to make connections between different

---

[*]Both authors contributed equally to this research.
[†]Corresponding author.

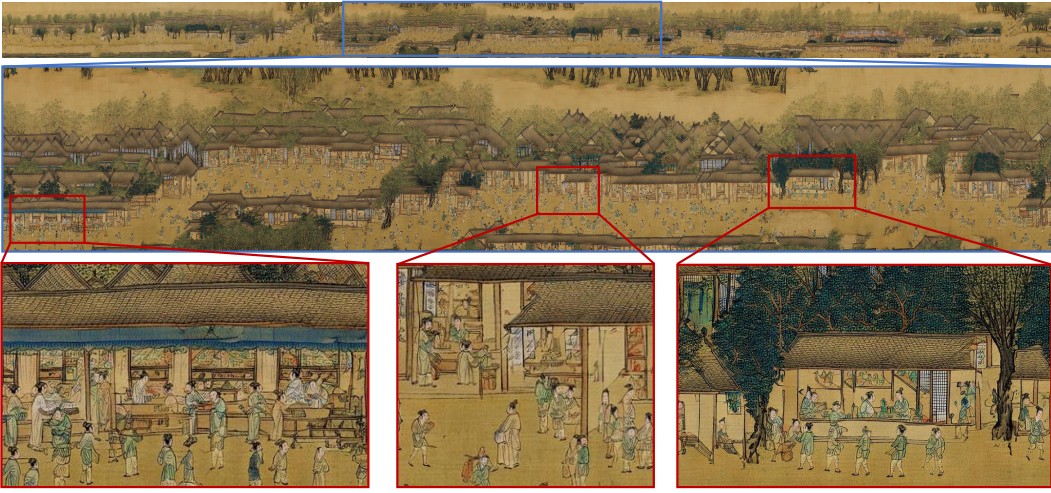

Figure 1: *The RiverSide of Qingming Festival* painted by our proposed NUWA-Infinity model in a resolution of $2048 \times 38912$. Trained from scratch on the original painting.

patches during inference. However, since different patches are still trained separately, the sliding window only relieves the inconsistency between patches, failing to model the global probability distribution of a large image or video.

We propose NUWA-Infinity, a patch-level[3] *"render-and-optimize"* strategy for infinite visual synthesis. When training on a large image, a rendering model auto-repressively predicts each patch based on its contexts. Once a patch is predicted, the loss is calculated and the parameters are optimized immediately and its hidden states are saved as contexts for the next *"render-and-optimize"* process. This brings two benefits: ($i$) The autoregressive rendering process with information transmission between contexts provides an implicit global probabilistic distribution modeling of the whole image; ($ii$) The timely optimization process alleviates the optimization stress of the model and helps convergence. Further, an arbitrary direction modeling is proposed to help NUWA-Infinity auto-repressively learn in arbitrary directions.

Based on the above designs, NUWA-Infinity shows a strong synthesis ability on even extreme high-resolution images (see Fig. 1).

To sum up, the major contributions of this paper are:

- We propose NUWA-Infinity, a patch-level "render-and-optimize" strategy that models the global probabilistic distribution of a large image or video with fast convergence.
- We propose an arbitrary direction relative position embedding, which enables the free-direction extension of NUWA-Infinity.
- NUWA-Infinity shows surprisingly good performance on both image and video synthesis. We also conduct detailed ablations to show the trade-offs between different rendering model.

## 2   Related Work

**High-Resolution Visual Synthesis**   High-resolution visual synthesis is a hot topic recently. Most models, such as Cogview [7] and DALLE-2 [24], firstly generate a small image, and then gradually scale it up. However, these two separate steps will result in a transmission gap between the small and the high-resolution images. Since a small image only contains a few limited objects, the super-resolution model can only scale it up, but cannot generate new ones. This paper focuses on Infinite Visual Synthesis, which aims to generate images or videos of arbitrary size without super resolution.

**Infinite Visual Synthesis**   Existing GAN-based models mainly embed global latent vectors [32, 28, 9] or coordinates conditions [17, 30] into the latent space to model high-dimensional visual features,

---

[3]Since this paper focus on infinite visual synthesis, patch represents a large region, for example $256 \times 256$.

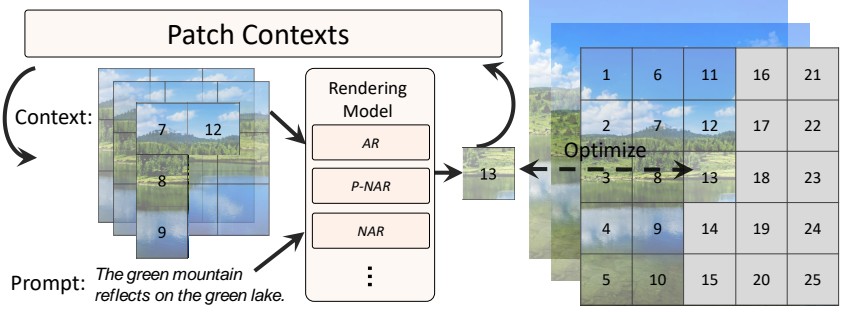

Figure 2: The overview of NUWA-Infinity model. When training on a visual data, a rendering model autoregressively predicts each patch based on its contexts. Once a patch is predicted, it is optimized immediately and its hidden states are saved as contexts for the next *"render-and-optimize"* process.

and then high-resolution images can be fast sampled from target distribution, but it usually faces the problem of complicated optimization, unstable training, and low diversity. Mask-Predict [3, 38, 2] as progressive non-autoregressive generation models designing complex sampling strategies on bidirectional masked language models, which generate several visual tokens in one step to speed up inference time, yet it needs to feed entire sequence into the bidirectional model which makes it impossible to scale up to a larger resolution. Auto-regressive models [8, 2] incorporate a sliding window to make a connection between different patches during inference. However, since different patches are still trained separately, the sliding window only relieves the inconsistency between patches, yet still failing to model the global probability distribution of data. This paper is also based on auto-regressive architectures, but aims to directly train a large image or video in an efficient way.

**Long-Range Sequence Modeling**   To efficiently model long-range sequences, on the one hand, the sparse computing is designed, like Axial Attention [14] and 3D Nearby Attention [36], they can reduce a lot of computation while maintaining visual quality; on the other hand, linking long sequences through memory information, memory-based models [6, 23, 5, 11] process the sequence in several segments and keep the sequence integrity with auxiliary historical memory. This paper proposes a "render-and-optmize" strategy and context pool to accelerate convergence and keep long-term memory for sequence modeling.

**Arbitrary Direction Extension**   Image out-painting and video prediction are considered as spatial and temporal extension respectively, therefore auto-regressive model can follow the orientation of different axes to extend images or videos naturally. However, since the direction of training AR is fixed, it can only be limited to extension in a certain direction. To support the extension in arbitrary directions, coordinate-based generative models [30, 17, 1, 27, 21] are proposed, They embed coordinate codes into latent space to generate a complete image, but this easily leads to repetition. Mask-Predict models [38, 2] benefit from a bidirectional language model, They iteratively mask and predict new regions to expand in any direction, while the context it relies on is only within the training window, resulting in a lack of the global consistency. Our approach is to change the training strategy of the AR model, the autoregressive direction during training covers the directions that can be extended, and this way allows us to use the AR model for arbitrary direction extension.

## 3   Method

Given an optional text prompt $y$, the infinite visual synthesis task aims to generate an arbitrary-size visual target $x$ (*e.g.*, a large image or a long video), *i.e.*, $\mathbb{P}(x|y)$. Since $x$ could be extremely large, current models [8] mainly try to split $x$ into several patches $\{p_1, p_2, ..., p_N\}$ and assume they are i.i.d during training, *i.e.*, $\prod_{i=1}^{N} \mathbb{P}(p_i|y)$, where $N$ is the number of patches, but this is a strong assumption. To consider the dependencies between patches, a simple idea is to model different patches in an auto-regressive manner, *i.e.*, $\prod_{i=1}^{N} \mathbb{P}(p_i|p_{<i}, y)$, but this requires the model to fully consider all the previous information. Hence the challenges turn to how to properly model $p_{<i}$ with affordable computing resources.

To address the above issues, we propose NUWA-Infinity, a patch-level *"render-and-optimize"* strategy that offers efficient training and inference whiling modeling $p_{<i}$ at the same time. We achieve this by incorporating context $c$ as a latent variable in Eq. 1:

$$\mathbb{P}\left(x|y\right) = \prod_{i=1}^{N} \mathbb{P}\left(p_i, c_i|p_{<i}, y\right) = \prod_{i=1}^{N} \mathbb{P}\left(p_i|c_i, y\right) \mathbb{P}\left(c_i|p_{<i}\right), \tag{1}$$

where $c_i$ should encode the information of $p_{<i}$. Eq. 1 consists of two components:

- **Rendering Model.** We define a plug-and-play rendering model that can be selected according to the balance of performance and speed. A rendering model $\mathbb{P}\left(p_i|c_i, y\right)$ that receives a visual context $c_i$ and an optional text prompt $y$, and then generates the current patch $p_i$.
- **Rendering Strategy.** We define a patch-by-patch rendering strategy that repeatedly invoke the rendering model. The patches rendered in the past $p_{<i}$ are used to generate a context $c_i$, which can be then used for the next rendering step.

Our approach uses a powerful visual rendering model $\mathbb{P}\left(p_i|c_i, y\right)$, though it can only generate a fixed-size patch $p_i$ each time, it considers the context $c_i$ of surrounding patches to ensure visual continuity and integrity. Through our designed patch-by-patch rendering approach, this rendering model will be repeatedly called to generate each patch; therefore, the modeling of $\mathbb{P}\left(p_i, c_i|p_{<i}, y\right)$ allows us to synthesize images or videos of any desired size.

## 3.1 Rendering Strategy

### 3.1.1 Arbitrary Direction Modeling

We encode raw visual pixels into discrete tokens by VQGAN [8], which significantly reduces the sequence length. Let $x \in \mathbb{R}^{h \times w \times f}$ be the visual tokens with height $h$, width $w$ and $f$ frames. Images are considered special videos with one frame. Firstly, we split $x$ into several non-overlapping patches $p \in \mathbb{R}^{h^p \times w^p \times f^p}$. Then we can get $(H \times W \times F)$ patches, where $H = \frac{h}{h^p}, W = \frac{w}{w^p}, F = \frac{f}{f^p}$. There are multiple ways to arrange these patches as an auto-regressive sequence. To enable the spatial extension in arbitrary direction, we define the generation direction $r$, including Down-Right-Forward, Down-Left-Forward, Right-Up-Forward and Right-Down-Forward. As shown in Fig 3 (b), we take Down-Right-Forward direction as an example. This direction denotes the patches are flattened from up to down as a first sort, from left to right as a second sort and from back to forward as the third sort. We consider that the coordinates of all patches are fixed, and different directions only change the time order of generation. Therefore, we define $T$ to convert coordinate $(i, j, k)$ into time order $n$ with a certain direction $r$, and $T^{-1}$ can deduce coordinates from time order $n$ and direction $r$.

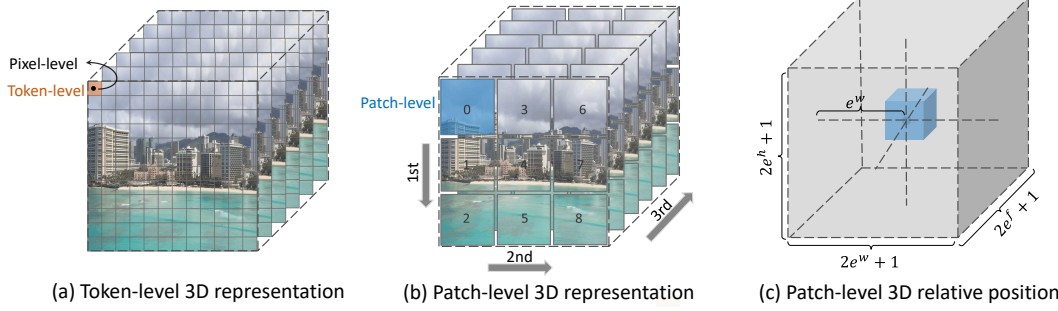

(a) Token-level 3D representation     (b) Patch-level 3D representation     (c) Patch-level 3D relative position

Figure 3: 3D representation for visual data. (a) visual tokens obtained by using VQGAN. (b) Each patch consists of several tokens, and the numbers on the patches indicate the time order of generation. (c) Relative position is based on the patches.

To be able to extend infinitely, how to efficiently encode a position of infinite size is crucial. Absolute position embedding [34] is unable to include all coordinates, while all coordinates can be represented by using a fixed number of relative position codes [15, 20]. Assuming that each patch only focuses on the information around it, we can define a receptive field expansion size as $(e^h, e^w, e^f)$ in Fig 3

(c), denoting the expansion size in the height, width and temporal axis respectively. Therefore we only need to create a learnable relative bias $\hat{B} \in \mathbb{R}^{(2e^h+1)(2e^w+1)(2e^f+1)}$, which can represent all position relationships between patches to represent infinite positional embeddings.

### 3.1.2 Nearby Context Pool

To make connections across patches, we propose a nearby context pool $q$. It can store the hidden states $m_{<i}$ of the previous patch $p_{<i}$. Then these hidden states will form the context $c_i$ as the rendering conditions of the patch $p_i$. Sec 3.1.1 mentioned that each patch will pay attention to the surrounding information within $(e^h, e^w, e^f)$, so the pool just needs to save the hidden states around each patch that have not yet been rendered. Although the pool does not store all hidden states, each hidden states is generated based on the previous hidden states, so this builds an information transfer chain to enable patches to obtain long-range contexts. We define three operations on this pool:

- **Select** the contexts of a patch at position $(i, j, k)$ from the pool $q$ with direction $r$.

$$c^{(i,j,k)} = q\left\{ m^{(i',j',k')} \,\middle|\, |i - i'| \leqslant e^h, |j - j'| \leqslant e^w, |k - k'| \leqslant e^t, T(i',j',k',r) < T(i,j,k,r) \right\} \quad (2)$$

- **Add** the hidden states of a patch at a position $(i, j, k)$ into the pool $q$:

$$q := q \cup \left\{ m^{(i,j,k)} \right\} \quad (3)$$

- **Remove** the expired hidden states from the pool $q$ at position $(i, j, k)$ with direction $r$.

$$q := q - q\left\{ m^{(i',j',k')} \,\middle|\, T(min(i' + e^h, H), min(j' + e^w, W), min(k' + e^f, F), r) \leqslant T(i,j,k,r) \right\}^4 \quad (4)$$

### 3.1.3 Training and Inference Strategy

We propose an infinite patch-by-patch rendering strategy, which is based on the nearby context pool, the training and inference method are defined below.

| **Algorithm 1:** Training Strategy | **Algorithm 2:** Inference Strategy |
|---|---|
| **Data:** images or videos $x$, optional text $y$ 
 **Result:** optimized rendering model 
 initial context pool $q \leftarrow \emptyset$ ; 
 $p[1..N] \leftarrow split(x)$; 
 $r \leftarrow$ sample training direction; 
 **for** *all $n$ from 1 to N* **do** 
      $i, j, k \leftarrow T^{-1}(n, r)$; 
      $c^{(i,j,k)} \leftarrow q.\textbf{Select}(i, j, k)$; 
      $\mathcal{L}^{(i,j,k)}, m^{(i,j,k)} \leftarrow \textbf{Render}(p^{(i,j,k)}, c^{(i,j,k)}, y)$; 
      $q.\textbf{Add}(m^{(i,j,k)})$; 
      $q.\textbf{Remove}()$; 
      *optimize* $\mathcal{L}^{(i,j,k)}$; 
 **end** | **Input:** optional text $y$, target size $s$ 
 **Output:** generated images or videos 
 initial context pool $q \leftarrow \emptyset$ ; 
 initial generation $g \leftarrow \emptyset$ ; 
 initial $n \leftarrow 1$; 
 **while** *not reached target size $s$* **do** 
      $c_{(i,j,k)} \leftarrow q.\textbf{Select}(n)$; 
      $p_n, m_n \leftarrow \textbf{Render}(c_n, y)$; 
      $q \leftarrow q.\textbf{Add}(m_n)$; 
      $q \leftarrow q.\textbf{Remove}()$; 
      $g \leftarrow g.\textbf{Add}(p_n)$; 
      $n \leftarrow n + 1$; 
 **end** 
 Return $merge(g)$; |

As summarized in Algorithm 1. Given a visual data $x$ and optional text $y$. Firstly, we split $x$ into several patches, and then we get $(H \times W \times F)$ patches, for each training batch we randomly sample a direction $r$, Finally, sliding rendering window to use *"render-and-optimize"*, the rendering model gets conditions and outputs loss of current patch, each loss of patch will be optimized separately, and then the updated parameters are used immediately for the next rendering.

---

[4]This equation only denotes Down-Right-Forward direction as an example.

## 3.2 Rendering Model

The Rendering model $\mathbb{P}(p_i|c_i, y)$ is a conditional generative model, it receives a context $c_i$ selected from the nearby context pool and a optional text prompt $y$, and then concentrate on generating the current patch $p_i$. We first provide the overall equation below:

$$z_{self}^l = SA(LN(q \leftarrow z^{l-1}, \ k \leftarrow [z^{l-1}, \ c^{l-1}] + b^{l-1}, \ v \leftarrow [z^{l-1}, c^{l-1}])) + z^{l-1} \ ,$$
$$z_{cross}^l = CA(LN(q \leftarrow z_{self}^l, \ k \leftarrow y^l, \ v \leftarrow y^l)) + z_{self}^l \ , \quad (5)$$
$$z^l = MLP(LN(z_{cross}^l)) + z_{cross}^l \ ,$$

where SA and CA are Self-Attention and Cross-Attention respectively, $LN$ is LaryerNorm and $MLP$ is linear layer. In SA, $z^{l-1}$ and $c^{l-1}$ denote the output and nearby context in $(l-1)$ layer, $b^{l-1}$ denotes the learnable relative position bias. we propose a pre-perceived relative position embedding (Pre-RPE), different from post-perceived relative position embedding (Post-RPE) [19], we add $b^{l-1}$ into the key. When producing attention-map, the location relationship between patches and contexts can be perceived in advance with Pre-RPE, instead of adjusting the already generated attention-map through Post-RPE. As a result, the SA outputs visual features with context. The CA produces cross-modal features $z_{cross}^l$ by interaction between visual outputs $z_{self}^l$ and text features $y^l$. The each layer output $z$ will be used as hidden states into the nearby context pool for next patch rendering.

We explore three rendering models:

**Autoregressive (AR)**: AR is a serial generation $\mathbb{P}(s_t^i|s_{<t}^i, c_i, y)$, where each discrete token $s_t^i$ will be predicted autoregressively. The input of first layer in the AR model is a sequence, which includes a BOS token and discrete visual tokens by VQGAN, we use the learnable axial absolute position encoding for this visual sequences. Tokens are predicted sequentially based on previous tokens:

$$\mathcal{L}_{patch}^{ar} = -\mathbb{E}_{t \in [1, h^p w^p f^p]} \ [log \ \mathbb{P}_\theta(s_t|s_{<t}, c, y)] \quad (6)$$

**Non-Autoregressive (NAR)** : The tokens in NAR are independent of each other, and they only depend on the input context $c_i$ and text prompt $y$. Therefore, $P(p_i|c_i, y)$ where patch $p_i$ can be generated directly in parallel. There are no serial dependencies between tokens in NAR, the sequence will be generated in parallel, and this greatly accelerates sampling speed. The first layer input of NAR is only an initialization sequence with position encoding, and do not require the BOS. In addition, since the visual input during training is position embeddings without the target tokens, we need to run NAR twice, the first time is used to optimize the model by $\mathcal{L}_{patch}^{nar}$ 7 , the second running is to get the each layer output, which will be saved as cache into nearby context pool.

$$\mathcal{L}_{patch}^{nar} = -\mathbb{E}_{t \in [1, h^p w^p f^p]} \ [log \ \mathbb{P}_\theta(s_t|, c, y)] \quad (7)$$

**Progressive Non-Autoregressive (P-NAR)** : To improve the generation performance of NAR. In P-NAR $P(p_i|\overline{p_i}, c_i, y)$ the previously generated sequence $\overline{p_i}$ will be iteratively optimized. P-NAR can be considered a Mask-Predict [38, 3, 2], it predicts all target tokens when given a fully-masked sequence at the first iteration, and then iteratively re-mask and re-predict a subset of tokens with low probability scores for a constant number of iterations. Follwing MaskGIT [2], We use the cosine mask scheduling function $\gamma \sim Cosine(\mathcal{U}(0,1))$, then masked tokens $s_m$ sampling from $\mathcal{D} \in \gamma \cdot (h^p w^p f^p)$. Eventually, through the non-masked tokens $s_{\overline{m}}$, it predicts the masked tokens:

$$\mathcal{L}_{patch}^{p\text{-}nar} = -\mathbb{E}_{m \in \mathcal{D}} \ [log \ \mathbb{P}_\theta(s_m|s_{\overline{m}}, c, y)] \quad (8)$$

## 4 Experiments

### 4.1 Experiment Setup

**Datasets.** For image synthesis, we trained unconditional generation model on the LHQ [30], which consists of 90k high-resolution ($\geqslant 1024^2$) nature landsacapes. In addition to support text prompt, we added a caption for each image of LHQ to create a new dataset called LHQC, where 85k as training data and 5k as test data. For video synthesis, we downloaded 120k high-resolution videos from pexels website and ran a pretrained Mask R-CNN [12] to remove the videos that likely contain objects on them. As a result we keeped 40k high-quality videos called LHQ-V.

**Metrics.** For image synthesis, we use Inception Score (IS) [26] and CLIP Similarity Score (CLIP-SIM) [22] to evaluate the sample diversity and semantic consistency between images and text. In

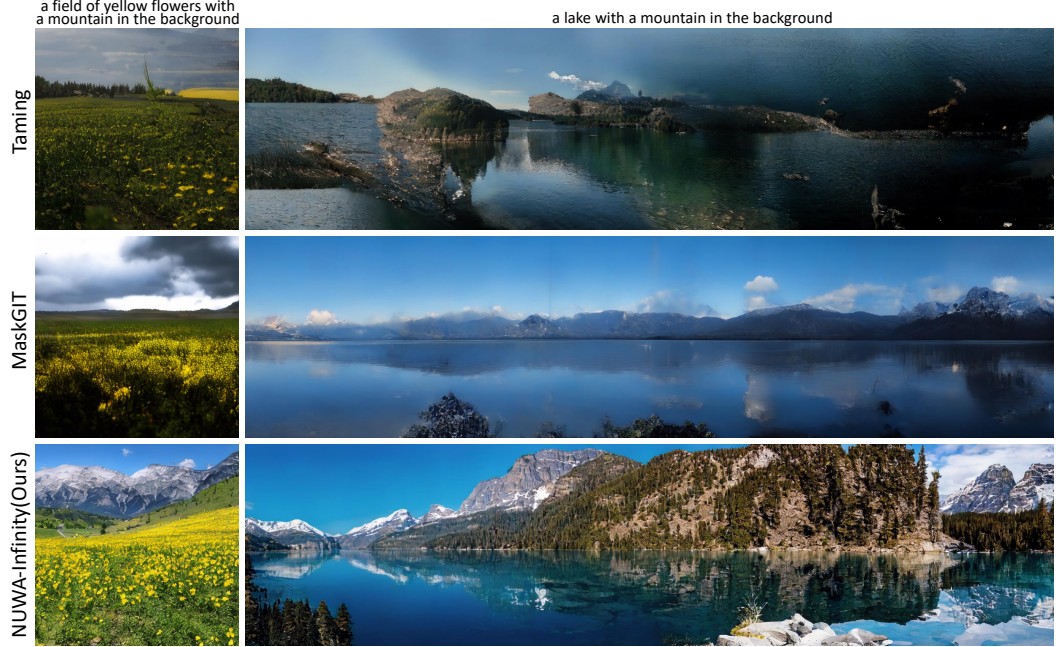

Figure 4: High Resolution text-to-image, the left is 1024×1024, the right is 1024×4096.

addition, we propose a **Block-FID**, which splits the large image into blocks to calculate Fréchet Inception Distance [13], it can avoid the downsampling caused by resizing images and work for infinite-size synthesis. Since rendering windows and blocks (256×256 in our experiments) maybe overlap, when measuring our model we will generate larger images, and move half of the block size. For video metrics, because the resize operation has little effect on the motion of the video, we use Fréchet Video Distance (FVD) [33] directly to evaluate the quality of the generated videos.

**Implementation Details.** During training, images are cropped into 1024×1024 and videos are cut into 1024×1024×5 with 5fps, then, they will be encoded into discrete tokens using the VQGAN model with a compression rate of 16 and a codebook of 16384. In Sec. 3.2, the rendering size of the three models is 256×256. In Sec. 3.1, based on the nearby sparsity, we set $(e^h, e^w, e^f) = (2, 2, 0)$ for images and $(e^h, e^w, e^f) = (1, 1, 3)$ for videos. We train the model using an Adam optimizer [16] with learning rate of 1e-4, a batch size of 256, and warm-up 5% of total 50 epochs.

## 4.2   Evaluation on Visual Synthesis

**High Resolution Text-to-Image**   We evaluate NUWA-Infinity on high resolution text-to-image task with LHQC dataset in Tab. 1. Under the 1024×1024 resolution same as the size of the training dataset, our model exceeds other methods with Block-FID of 9.71, as well as better text-image semantic consistency of 0.2807 and more diversity of 4.98. When generating larger images such as 1024×4096 (×4) resolution, the performance of MaskGIT [2] decreases rapidly due to limited Window, but our model can still maintain higher visual quality with Block-FID of 15.65.

| Method | Block-FID↓ | Block-FID(×4)↓ | IS↑ | CLIP-SIM↑ |
|---|---|---|---|---|
| Taming [8] | 38.89 | 46.37 | 4.58 | 0.2662 |
| MaskGIT [2] | 24.33 | 45.76 | 4.61 | 0.2754 |
| Our (AR) | **9.71** | **15.65** | **4.98** | **0.2807** |

Table 1: High resolution text-to-image synthesis on LHQC, the size of sample is 1024×1024 by default, ×4 means 1024×4096.

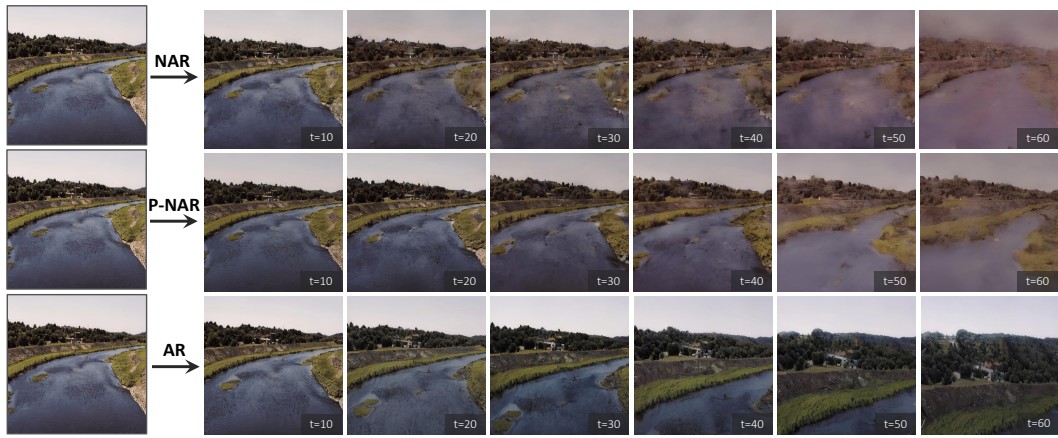

Figure 5: Long-duration video prediction, generating 60 frames from 1 frame.

**High Resolution Image-to-Video**  Since our framework generates patches autoregressively, and it can easily expand from image synthesis to video synthesis. We compare the performance of three different rendering models for video prediction in Tab. 2 and Fig. 5.

| Method | FVD↓ |
|---|---|
| StyleGAN-V[31] | 143.76 |
| Our (NAR) | 368.16 |
| Our (P-NAR) | 165.39 |
| Our (AR) | **62.57** |

Table 2: Video prediction on LHQ-V, the size of sample is 1024×1024×8.

| Method | Block-FID↓ | | | |
|---|---|---|---|---|
| | Right Extend ⇒ | Left Extend ⇐ | Down Extend ⇓ | Up Extend ⇑ |
| Taming [8] | 22.53 | N/A | 26.38 | N/A |
| MaskGIT [2] | 14.68 | 14.81 | 25.57 | 25.38 |
| InfinityGAN [17] w/o text | 17.93 | 19.76 | 24.62 | 23.59 |
| Our (AR) w/o text | **6.43** | **6.71** | 11.47 | 8.03 |
| Our (AR) w/ text | 6.45 | 6.72 | **9.84** | **7.43** |

Table 3: Arbitrary image extension on LHQC, calculating Block-FID between the extended area and the same area in the test set, not using full images. Taming and MaskGIT use text prompt by default.

**Arbitrary Image Extension**  Tab. 3 and Fig. 6 demonstrates that our proposed approach achieves state-of-the-art performance on arbitrary image extension task. Traditional AR models like Taming [8] only generate samples along a certain order and can not achieve image outpainting in arbitrary direction. MaskGIT benefits from a bidirectional masked language model, which can add a new mask area in any direction and predict it to extend images. However, since MaskGIT is inconsistent between the mask area of training images and extending images, and this damage its performance. In addition, we found that the text prompt does not help when extending the image left or right, while the text prompt brings performance improvements when extending up or down. We think it is because upper or lower half of the image contains less information, while the left or right half has more visual semantic hints.

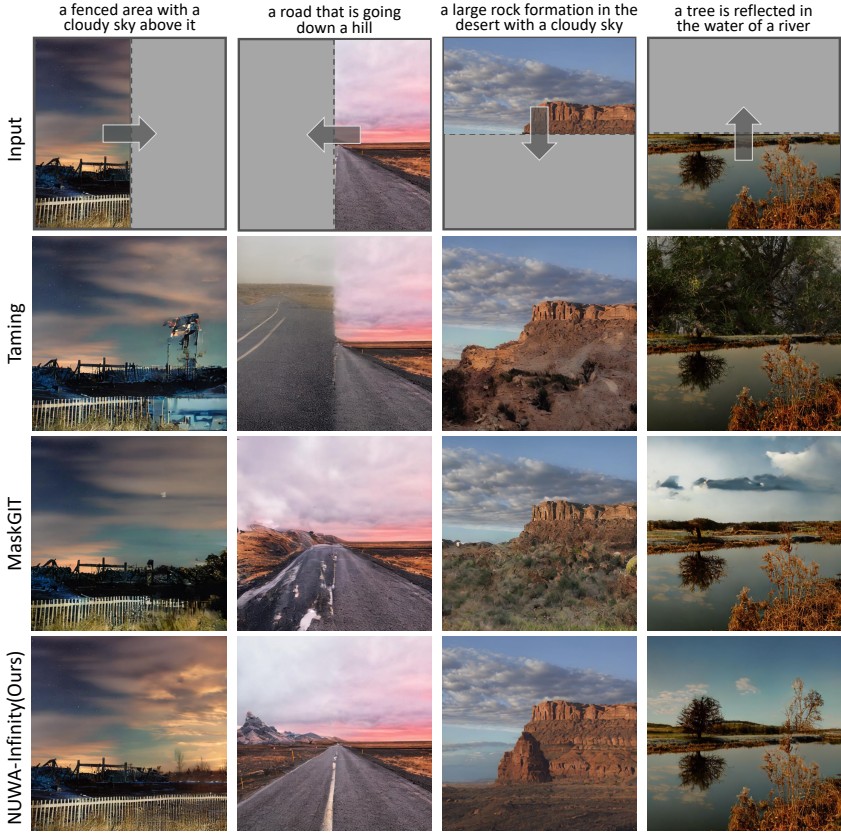

Figure 6: Arbitrary direction image extension, the input is 1024×512.

## 4.3 Ablation Studies

**Rendering model**  Tab. 4a shows that the AR rendering model has the best performance but sacrificed the generation speed. P-NAR optimizes the results of NAR multiple times and achieve good performance at a not bad speed. Scaling up the model in Tab. 4b and using Pre-RPE in Tab. 4c can improve the model performance. We noticed in Fig. 7a that the rendering size has a great impact on visual quality and smaller rendering size will lead to worse performance, especially for 1024×4096 (x4) resolution generation. It is because of too small patches as low-level visual features, and it is not conducive to model learning the global consistency. When rendering size is equal to 256, this phenomenon will disappear, but if size continues to expand, it easily leads to out of memory.

**Training way**  Tab. 4e shows that our synthesis-and-optimize training strategy can accelerate convergence from 70 epochs to 50 epochs. If the model accumulates the gradient of the full images to optimize, it is not conducive to sharing the parameters between each patch, so that the rendering model is too concerned about the global layout without focusing connection between each patch. In addition, if each patch is optimized by itself and this can provide more data distribution. The success of synthesis-and-optimize strategy also benefits from the long-term connections between patches by context transfer. Tab. 4d proves that context transfer will bring huge improvements, where no transfer means the cache generated by each patch only contains its own information.

**Nearby extension**  Although implicit context transfer can make the cross-patch connection distance farther, explicitly increasing expansion size in nearby context pool will bring more stable connections. Fig. 7b shows the impact of spatial expansion size on image generation, as $e^h$ and $e^w$ increase, performance will quickly improve until size is equal to 2, but FLOPs will continue to grow. Fig. 7c presents the impact of temporal expansion size on video prediction. Without considering context transfer, when $e^f$ is equal to 1, the model can obtain the previous one static frame. When $e^f$ is equal to 2, the model can obtain the action between the two static frames, but when $e^f$ reaches 3, the model

| Render model | Block-FID↓ | Block-FID(× 4)↓ | CLIP-SIM↑ | Inference Speed↑ |
|---|---|---|---|---|
| NAR | 92.34 | 98.67 | 0.2451 | **95×** |
| P-NAR | 19.86 | 38.59 | 0.2726 | 15× |
| AR | **10.05** | **17.78** | **0.2753** | 1× |

(a) **Render model.**

| Parameters | Depth | Dim | Block-FID↓ |
|---|---|---|---|
| 202M (Base) | 16 | 768 | 10.05 |
| 809M (Large) | 24 | 1280 | **9.71** |

(b) **Model size.**

| RPE | Block-FID↓ | Block-FID(×4) ↓ |
|---|---|---|
| Pre | **10.05** | **17.78** |
| Post | 10.47 | 18.89 |

(c) **Relative position embedding.**

| Context | Block-FID↓ | Block-FID(×4) ↓ |
|---|---|---|
| w/o transfer | 15.8 | 38.32 |
| w/ transfer | **10.21** | **23.62** |

(d) **Context transfer.**

| Loss | Block-FID↓ | Convergence epoch↓ |
|---|---|---|
| Patch | **10.05** | **50** |
| Full | 11.62 | 70 |

(e) **Loss region.**

Table 4: Ablation experiments with text-to-image on LHQC, the experiments use base model with layers of 16 and dim of 768 except (b). Default setting are marked in gray.

can obtain the movement between two dynamic visual features. So starts from $e^f = 3$, the growth of the performance is not obvious.

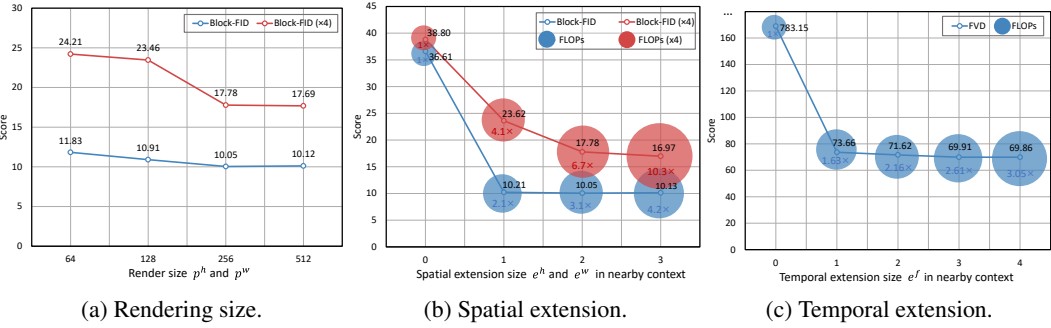

(a) Rendering size.  (b) Spatial extension.  (c) Temporal extension.

Figure 7: Ablation results on rendering size and nearby expansion size, (a) and (b) are based on text-to-image synthesis, (c) is based on video prediction with 8 frames. Note that they all use the base model with layers of 16 and dim of 768.

## 5  Conclusion

In this work, we propose NUWA-Infinity, a patch-level *"render-and-optimize"* strategy for infinite visual synthesis. In training stage, visual data are spitied into non-overlapping patches with different time order, then a rendering model autorepressively predicts each patch based on its contexts, the patch is optimized at each time step, and the optimized parameters are immediately used in the next rendering. This strategy allows us to 1) generate images and videos of infinite size 2) visual extension in any direction 3) information transfer brings long-term memory 4) accelerate model convergence.

## 6  Acknowledgements

The paper is supported by the National Key Research and Development Project (Grant No.2020AAA0106600).

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
