# A  Architecture and Hyperparameters

We proposed NUWA-Infinity is a two-stage approach. In the first stage, we encode raw visual pixels $1024 \times 1024$ into discrete tokens $64 \times 64$ by VQGAN with a compression ratio of 16. In second stage, we train a rendering model based on transformer architecture.

| Setting | NUWA-Infinity (Text-to-Image) | NUWA-Infinity-V (Image-to-Video) |
|---|---|---|
| VQGAN codebook | 16384 | 16384 |
| VQGAN dimension | 256 | 256 |
| VQGAN compression ratio | 16 | 16 |
| Transformer layer number | 24 | 24 |
| Transformer hidden dimension | 1280 | 1280 |
| Transformer head number | 20 | 20 |
| Transformer self-attention | ✓ | ✓ |
| Transformer cross-attention | ✓ | ✗ |
| Patch size | 256 | 256 |
| Nearby expansion size | $(2, 2, 0)$ | $(1, 1, 3)$ |
| Dataset | LHQC | LHQ-V |
| Training number | 85K | 38K |
| Test number | 5K | 2K |
| Training epoch | 50 | 50 |
| Visual input size | $1024 \times 1024 \times 1$ | $1024 \times 1024 \times 5$ |
| Text input size | 77 | N/A |
| Batch size | 256 | 256 |
| Learning rate | 1e-4 | 1e-4 |
| Warmup ratio | 5% | 5% |

Table 1: Implementation details for the large model

As shown in Tab. 1. NUWA-Infinity by default refers to the model of text-to-image synthesis. Texts will be encoded into the tensors of $77 \times 512$ size by pretrained text encoder of CLIP, and they are fed into cross-attention as key and value to interact with visual features. We use three data augmentations on the images including RandomResizedCrop, RandomHorizontalFlip and ColorJitter. The image is only spatially expanded, so we choose $(e^h, e^w, e^f) = (2, 2, 0)$ expandsion size for image synthesis. NUWA-Infinity-V refers to the model of image-to-video synthesis. Since it mainly expands the videos in the temporal axis, we choose a smaller spatial receptive field $(e^h, e^w) = (1, 1)$ but a larger temporal receptive field $e^f = 3$. Each $1024 \times 1024 \times 5$ clip is cropped from video with 5fps, and they also apply the three data augmentations mentioned above.

For training, we employ an Adam optimizer for 50 epochs using 5% of linear warm-up to a peak learning rate of 1e-4 and a linear decay learning rate scheduler. For inference, we use different sampling strategies for each rendering model. AR models use top-k of 768, NAR models use gumbel sampling and P-NAR models use gumbel sampling with temperature annealing from 4.5 to 1.0.

# B  Computational Cost

Sparsity can greatly reduce the computational cost. Supposing that each training data contains $(H \times W \times F)$ patches, and defining nearby size $(e^h, e^w, e^f)$. Our proposed nearby context pool can make the computational complexity $O\left((HWF)\left(e^h e^w e^f\right)\right)$ instead of $O\left((HWF)^2\right)$ in Transformer. In addition, due to *"render-and-optimize"* strategy, the training input of each step is just the sequence of one patch, so our training complexity of one step is just $O\left(e^h e^w e^f\right)$ to support larger batch size.

Benefit from above design. We trained NUWA-Infinity (Large) for 1.25 days and NUWA-Infinity (Base) for 0.35 days on 64 A100 GPUs. Since video generation additionally needs to consider temporal information, we trained NUWA-Infinity-V (Large) for 3.7 days and NUWA-Infinity-V (Base) for 1 day on 64 A100 GPUs.

## C   Details for Arbitrary Direction

To support generation in arbitrary direction, we train NUWA-Infinity with different orders of patch-level sequence. Our default temporal direction is forward, and there are four basic spatial directions including right, left, down and up.

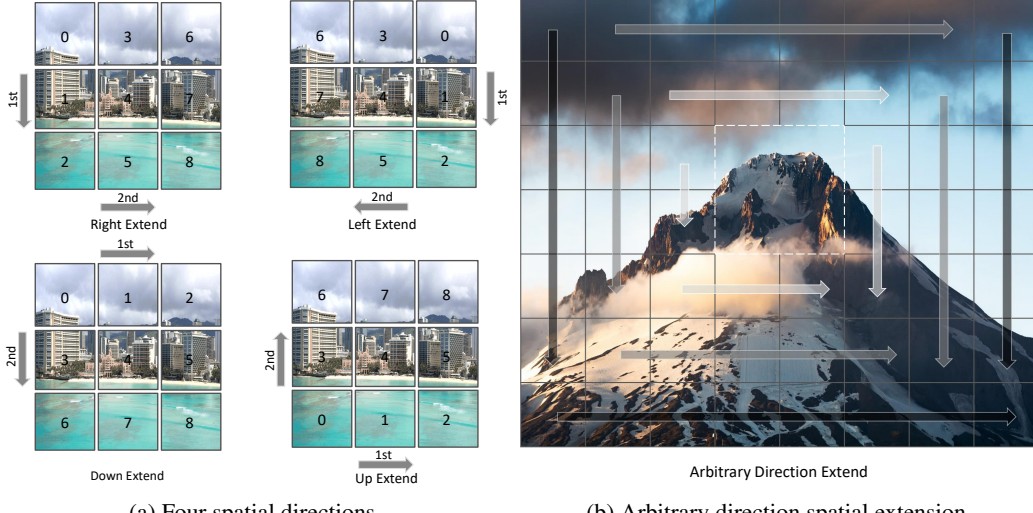

(a) Four spatial directions                    (b) Arbitrary direction spatial extension

Figure 1: Arbitrary spatial direction for generation. (a) shows the generation order of the basic four spatial directions including right, left, down and down. (b) describes the process of extending a small image to an image of any size.

We design four directions in Fig. 1a. During training we randomly sample one direction from these four directions to flatten patches. For example, up extension requires the rendering window to move to the right and then move up. In addition, we propose a loop circle way in Fig. 1b to iteratively use the basic four directions to extend images in arbitrary direction at the same time, and the default order of circle direction is left, down, right and then up.

## D   One-shot Training

One-shot training means the model trained from scratch with one training sample. The RiverSide of Qingming Festival on the homepage of this paper and Fig. 6 are the results of one-shot training. The *"render-and-optimize"* strategy can train each patch separately, it allows us make the most of every small area of a large image. Furthermore, to avoid generating the original image directly, we use label smoothing=0.15 and more data augmentations, additionally including Mirror, Rotation, Noise and Shear.

## E   Limitations

While NUWA-Infinity has significant performance in infinite visual synthesis, it still has the following limitations:

1. Our approach is a patch-level autoregressive generation model, the minimum granularity for generating images and videos is limited by the size of the patch, such as $128 \times 128$ and $256 \times 256$. Although we can generate larger results and crop them, this has some waste of computing resources and inconvenience in operation.

2. Although the context can be transferred implicitly through the pool, as the length of the generating sequence increases, long-distant information will inevitably be forgotten. Fig. 11 shows that the background color of the last frame has been inconsistent with the input frame.

3. When expanding images and videos spatially or temporally, we scan a given area serially to produce the context, not in parallel. This leads to an increase in the inference time.

a river flowing through a forest with mountains in the background

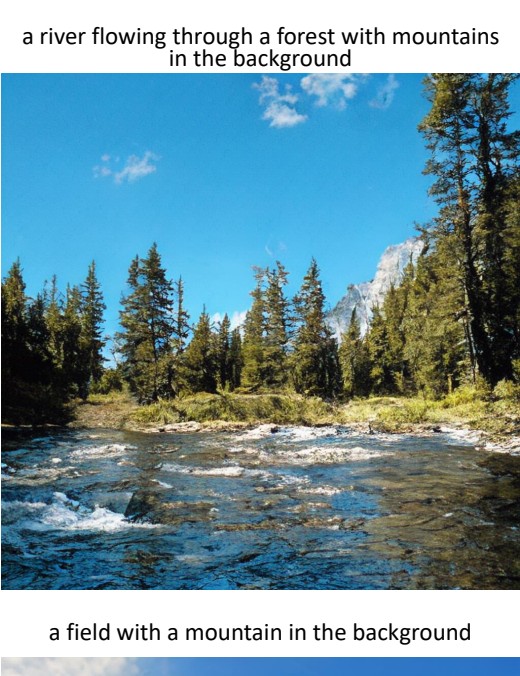

a road that is going down a hill

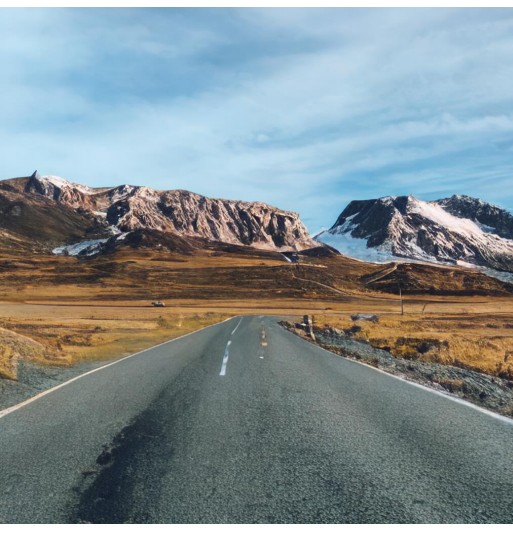

a sunset over the ocean with waves crashing on the shore

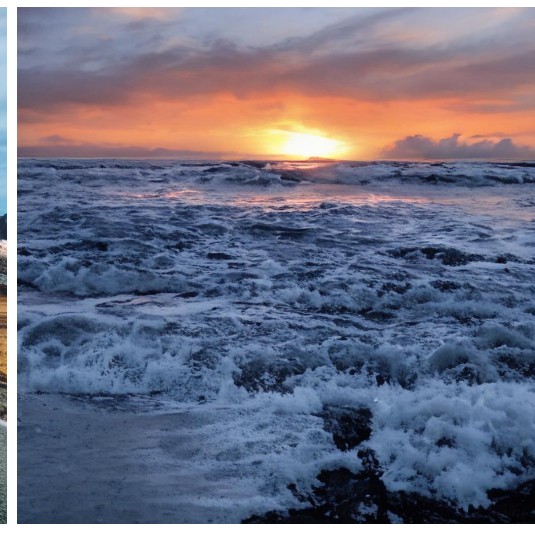

a field with a mountain in the background

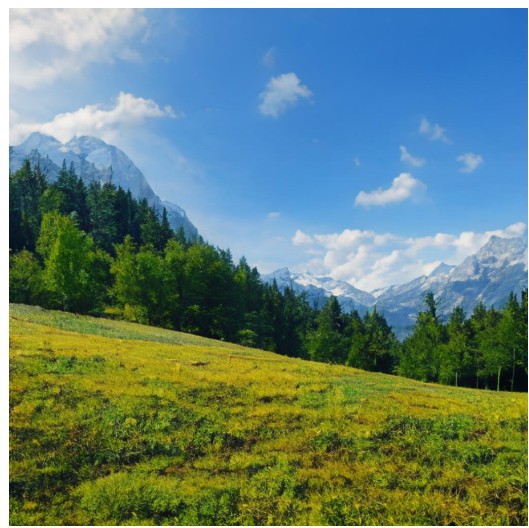

a field with a mountain in the background

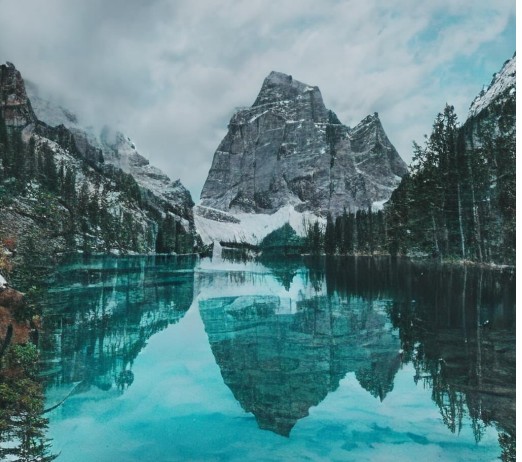

a mountain range with a cloudy sky at sunset

a tree with no leaves in the foreground

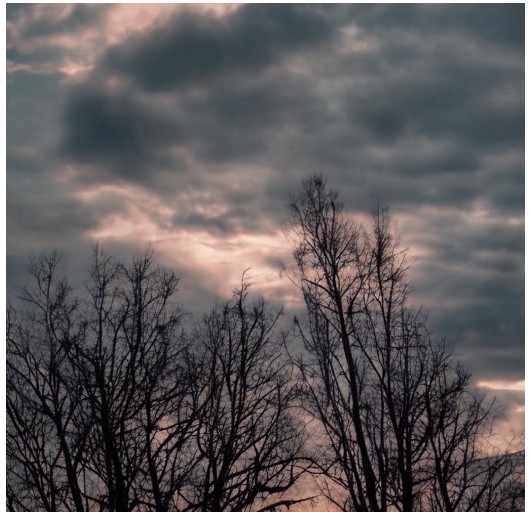

a group of trees that are silhouetted against a sunset

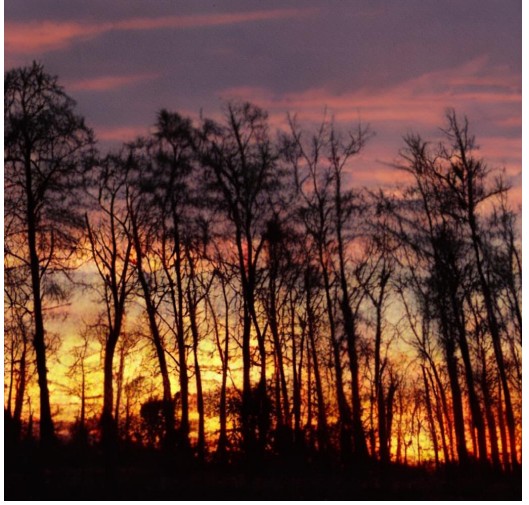

a city skyline with a red sky and clouds

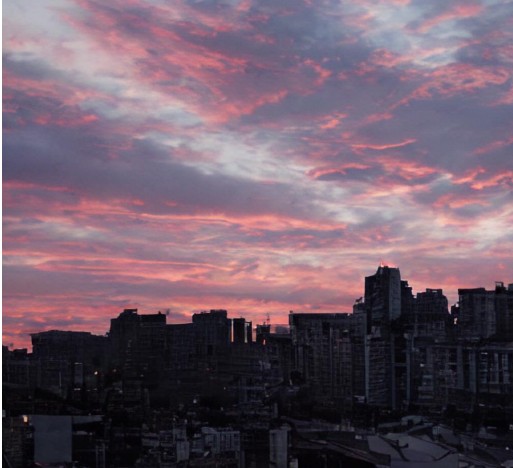

Figure 2: Text-to-Image 1024×1024 samples.

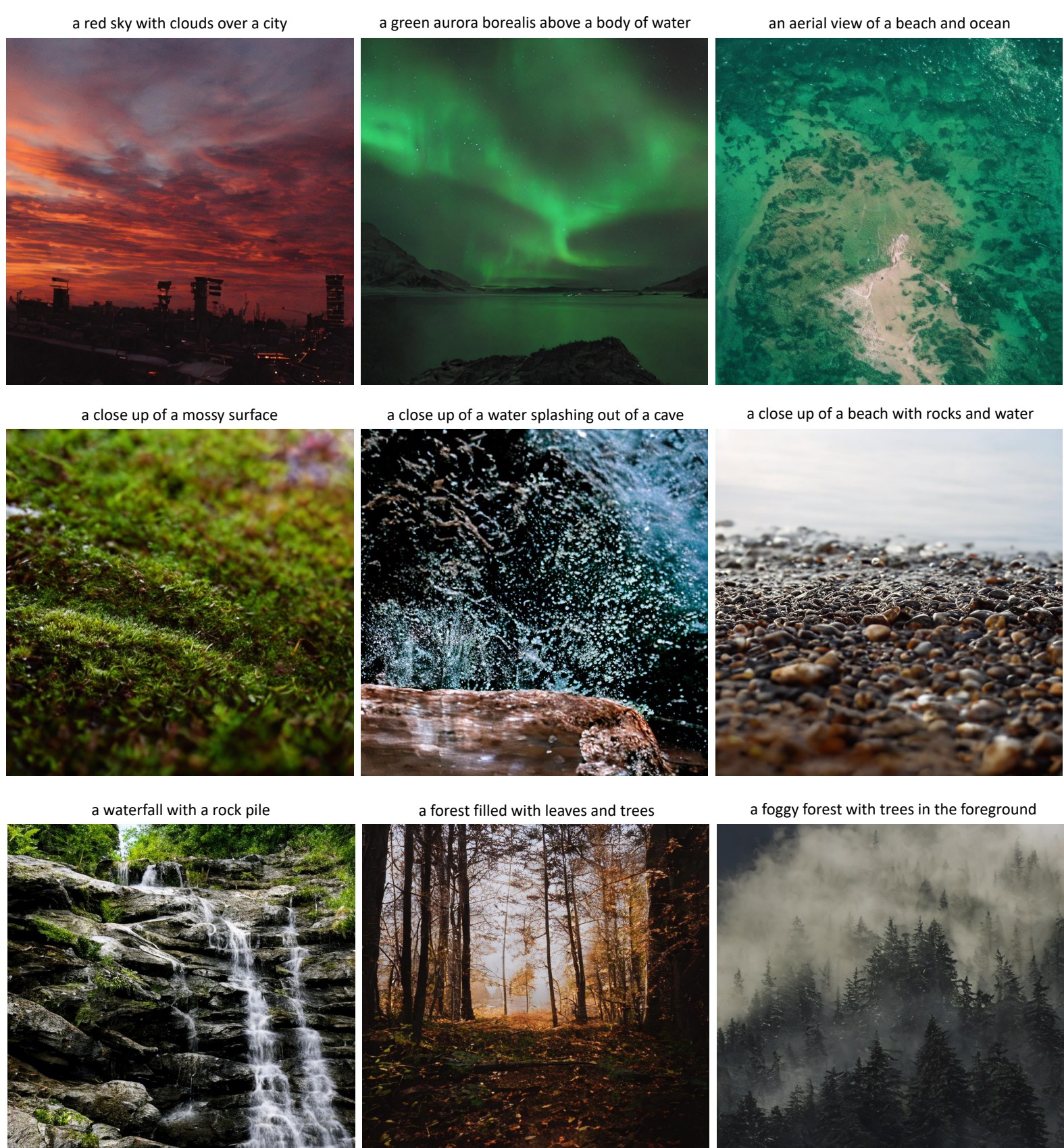

Figure 3: Text-to-Image 1024×1024 samples.

a field with a house and a cloudy sky

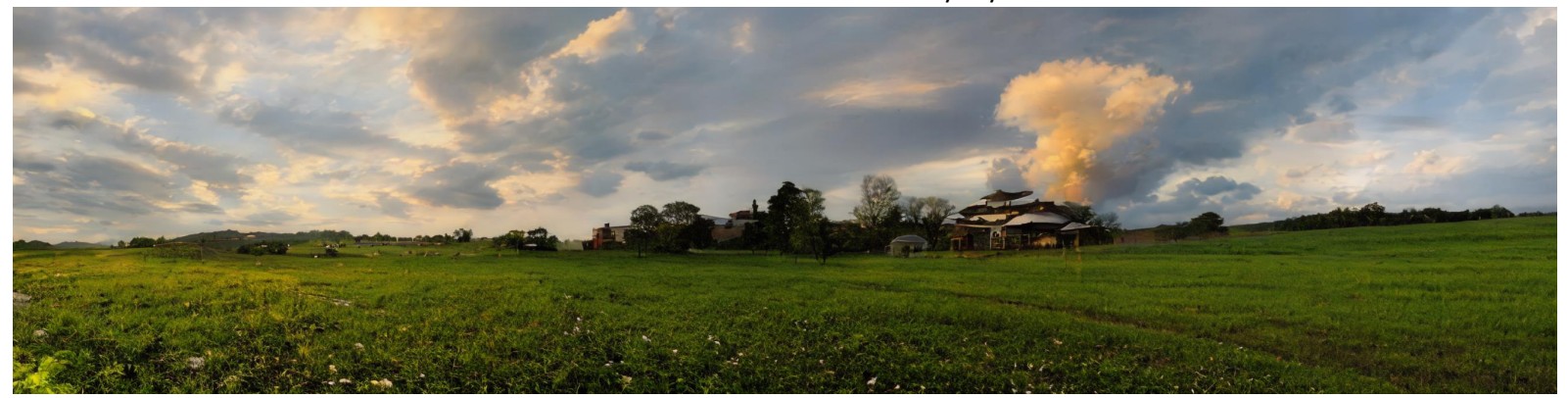

a large lake surrounded by green vegetation

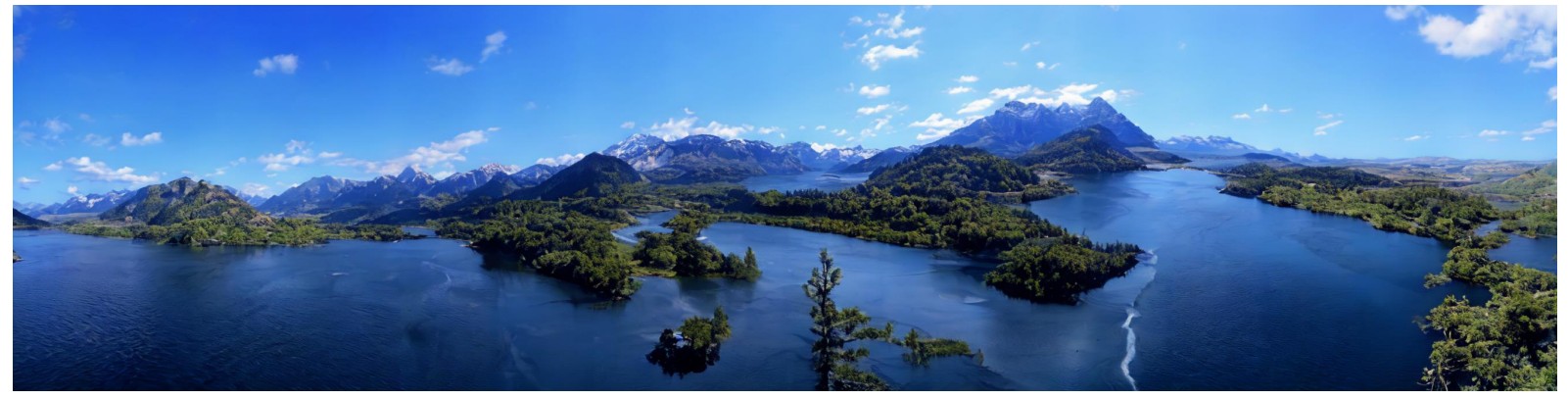

a desert landscape with trees and mountains in the background

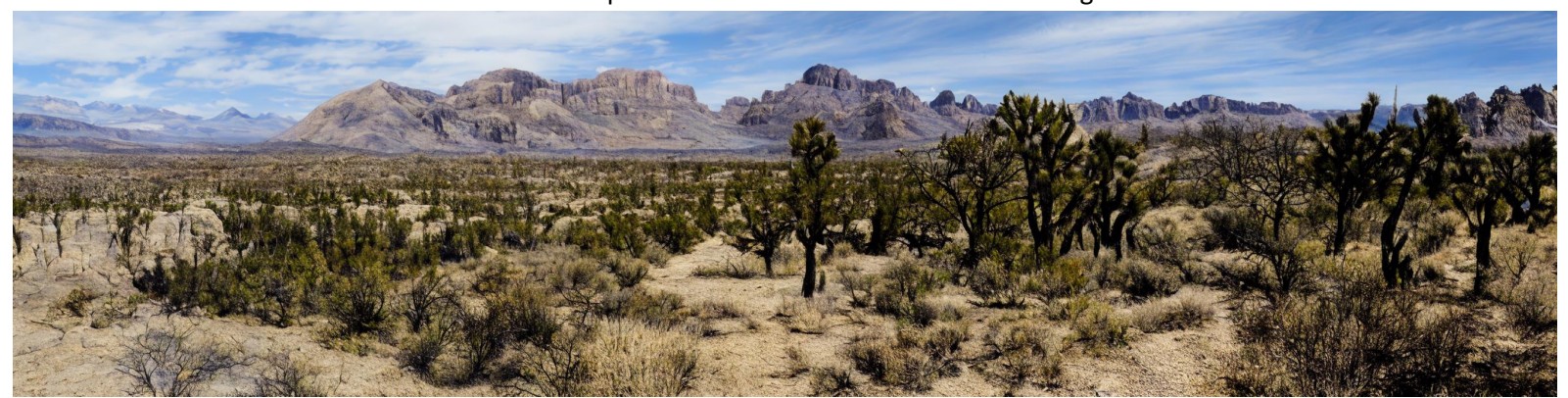

a cliff with a large canyon

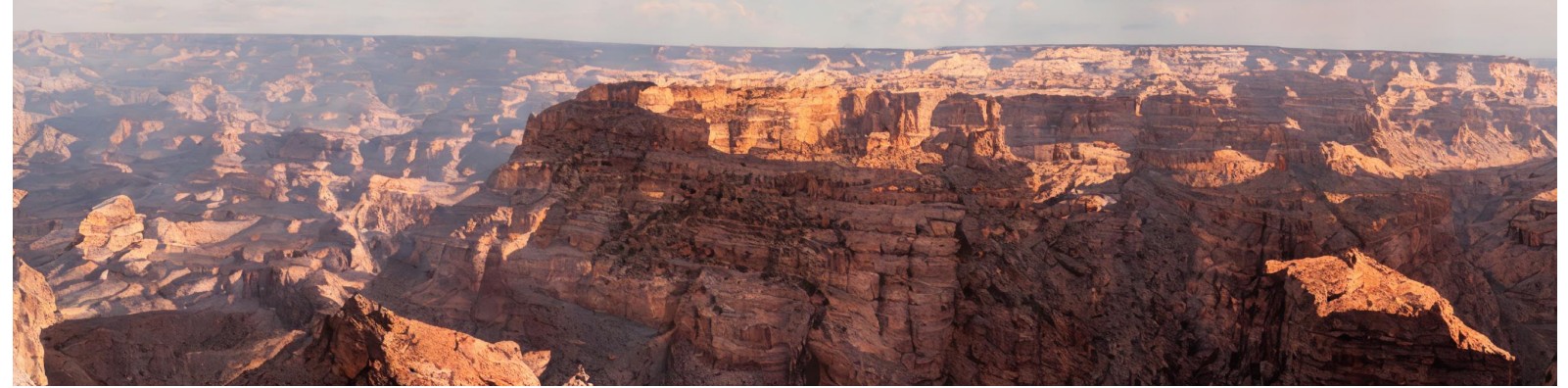

Figure 4: Text-to-Image 1024×4096 samples.

a mountain range with a cloudy sky at sunset

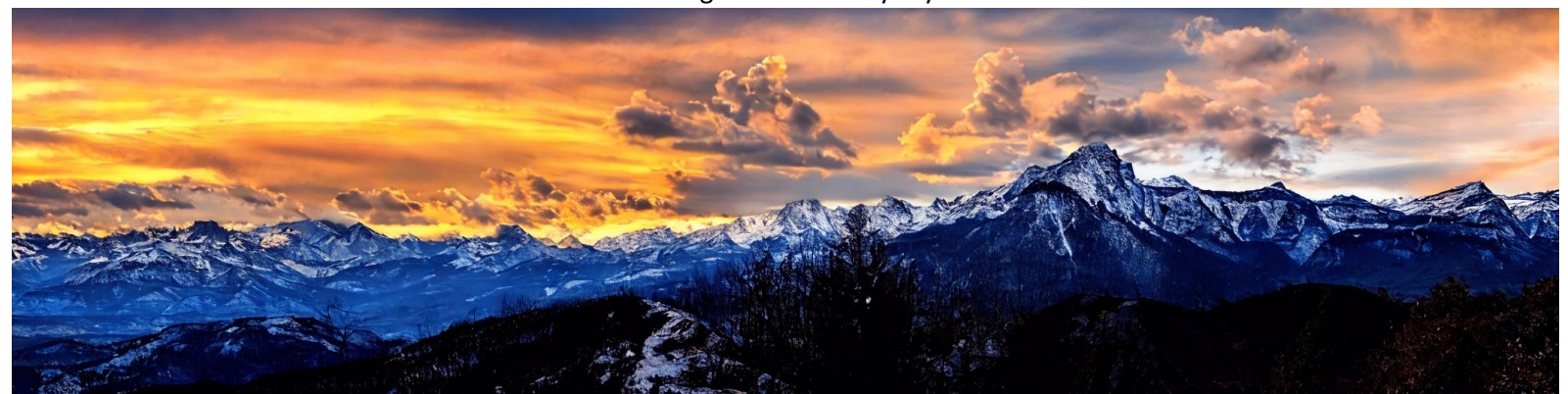

a red sky with clouds and trees silhouettes

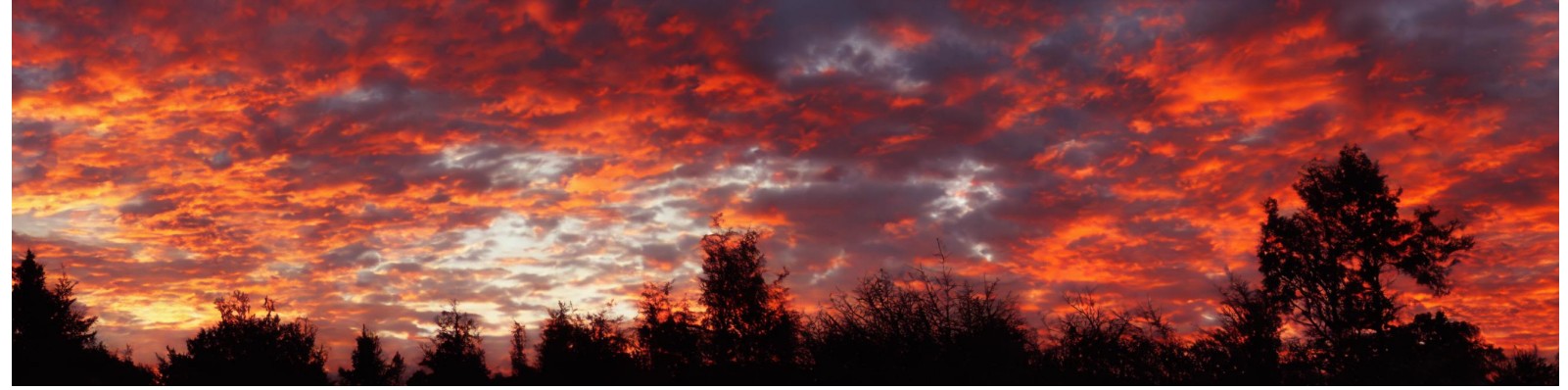

a waterfall is surrounded by rocks and trees

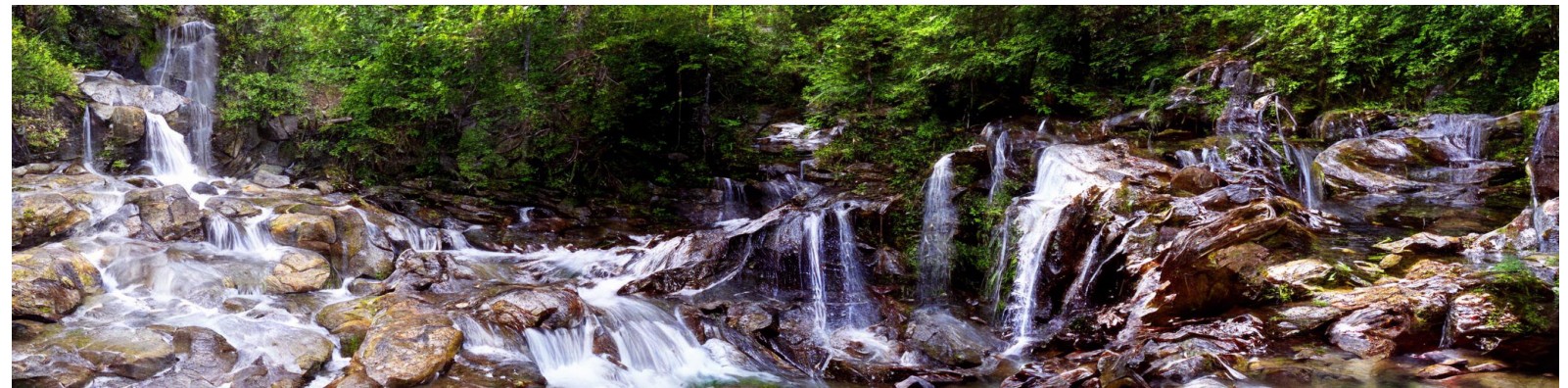

a city with a mountain in the background

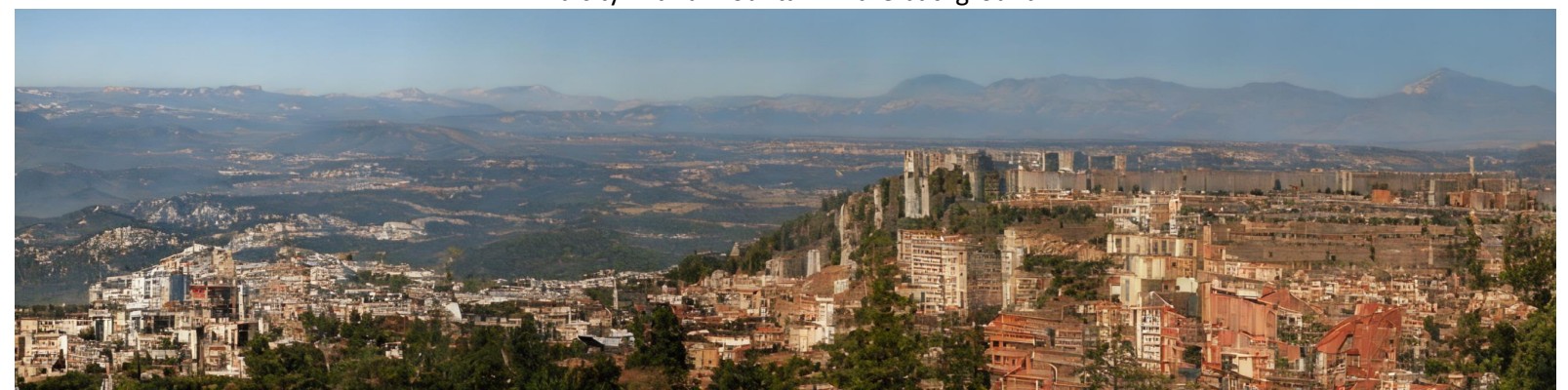

Figure 5: Text-to-Image 1024×4096 samples.

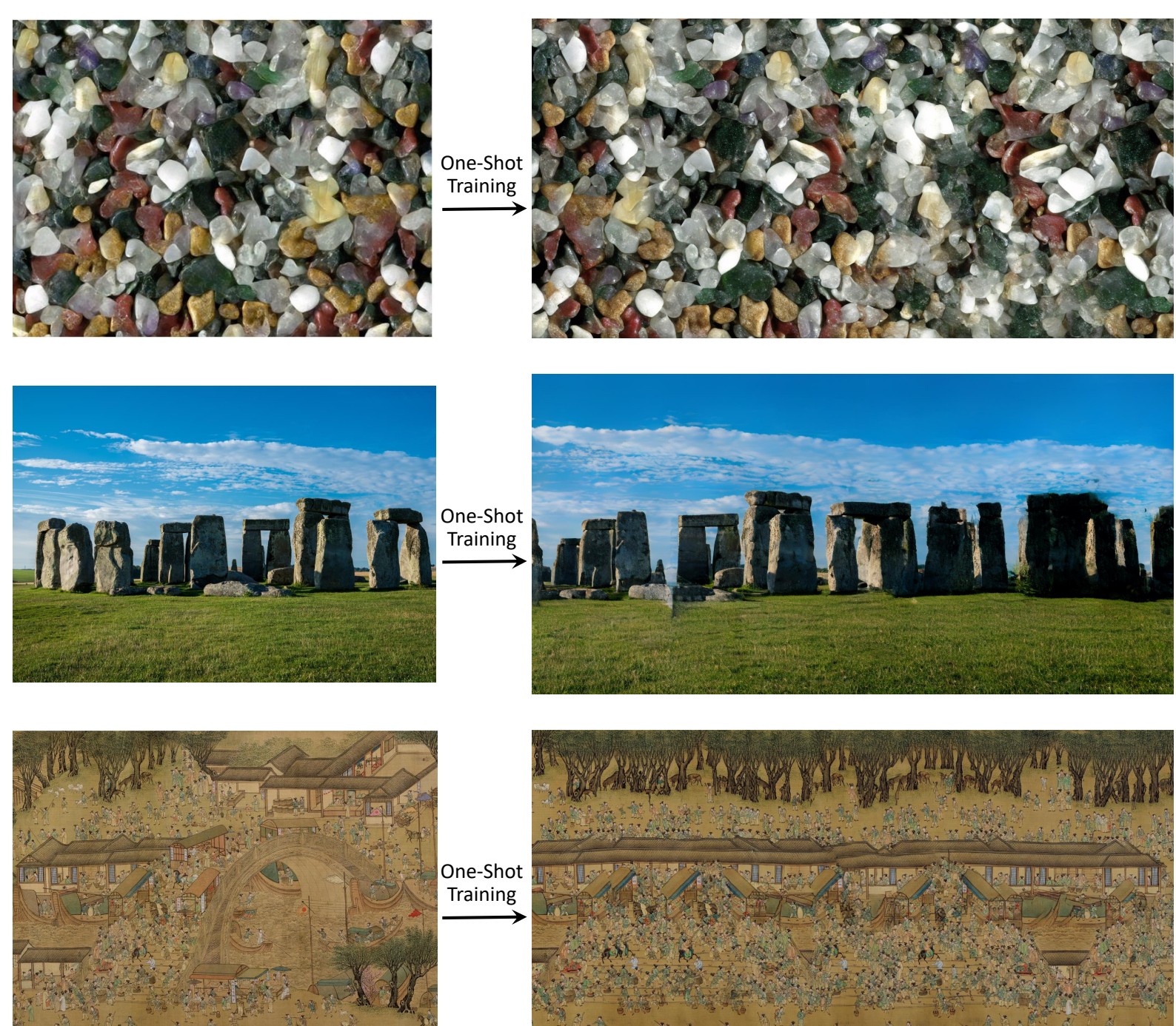

Figure 6: One-Shot Training 1024×2048 samples. Note that the last row shows only the partial image

a tree branch with green leaves and a forest in the background

a fallen tree over a waterfall in a forest

a house on a snowy hill under a starry sky

a city skyline with lights reflecting in the water

a cliff with a green field and a blue ocean

a waterfall with a green river in the background

a group of trees in a field with leaves

a dirt road that is on a hill

Figure 7: Image Extension 1024×2048 samples.

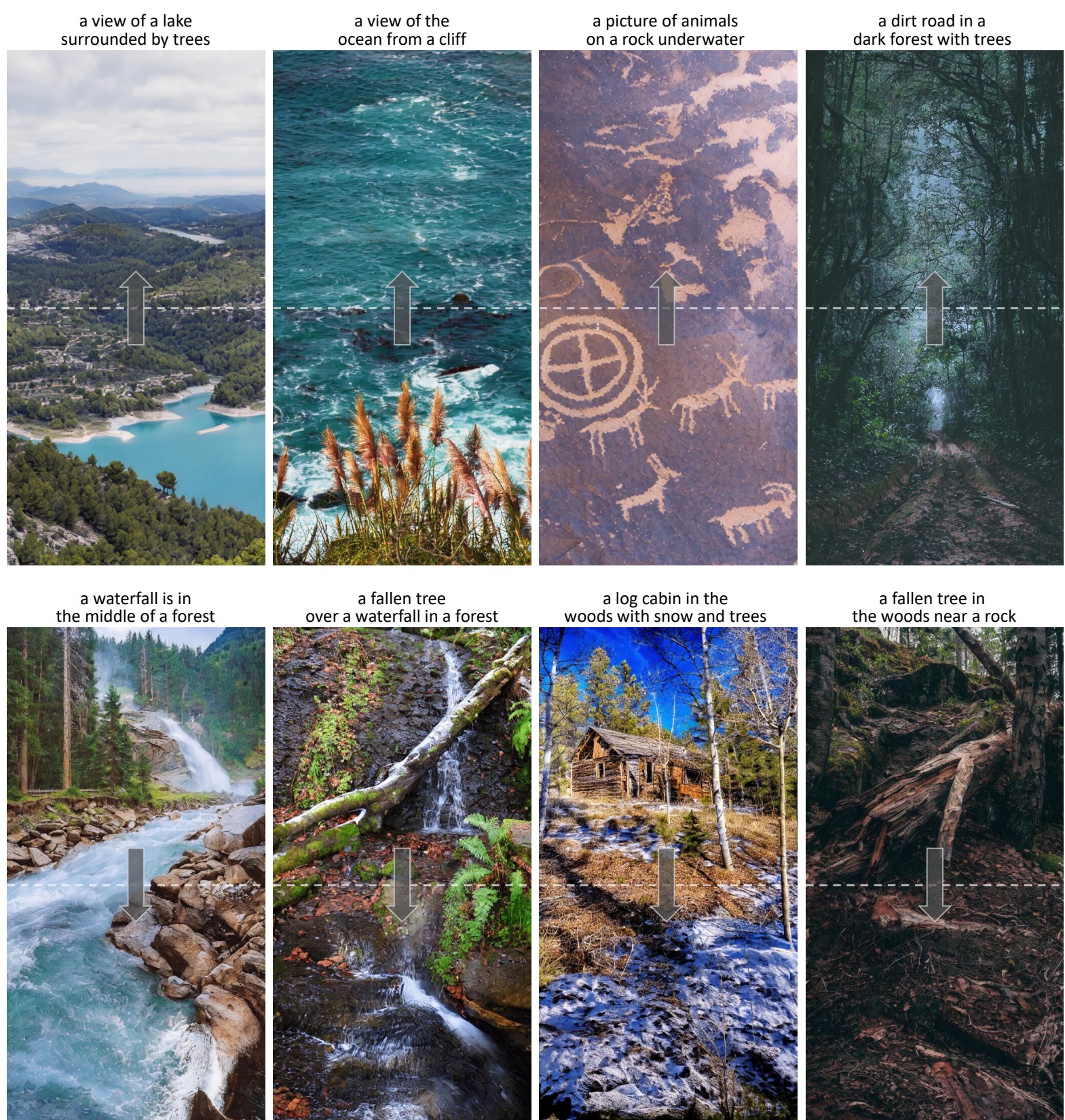

Figure 8: Image Extension 2048×1024 samples.

*The Starry Night*

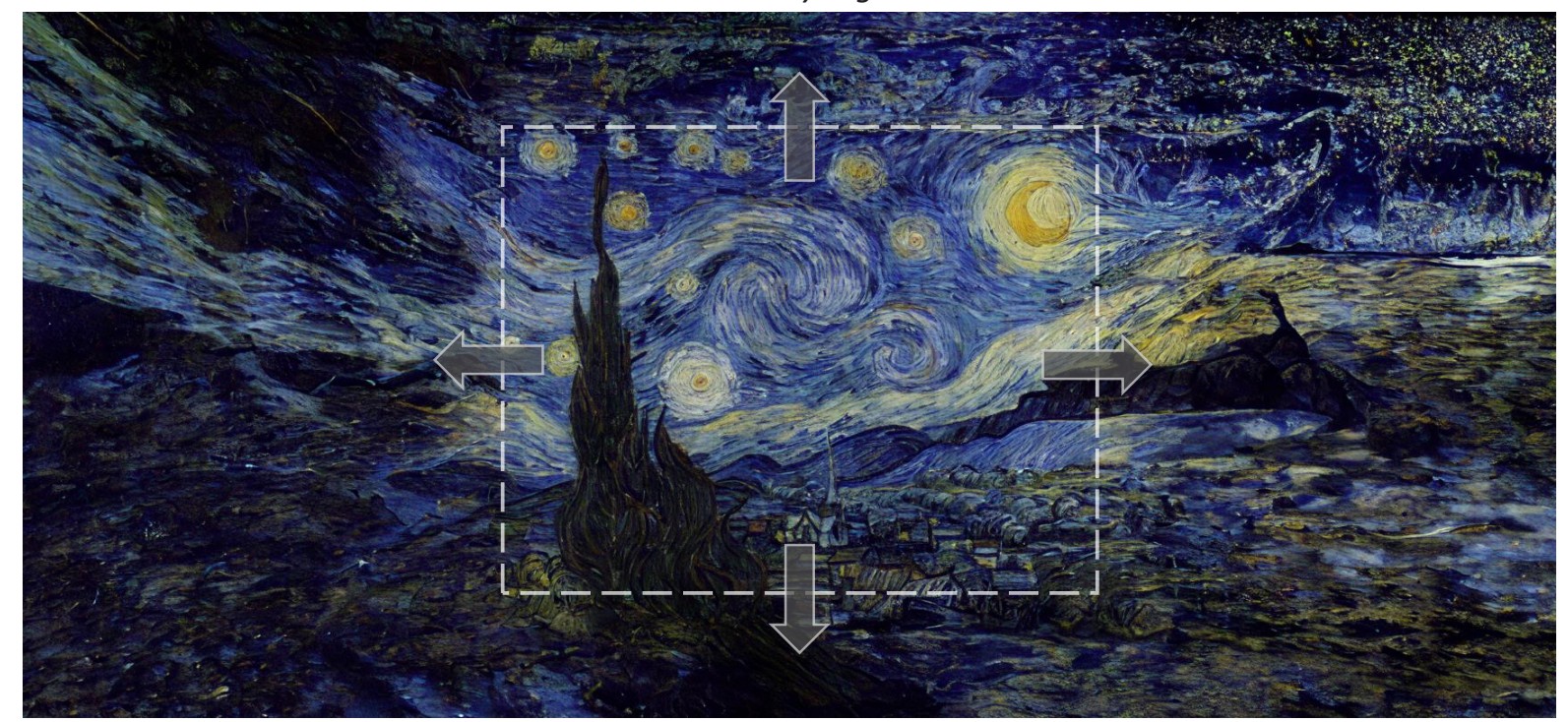

*The Gleaners*

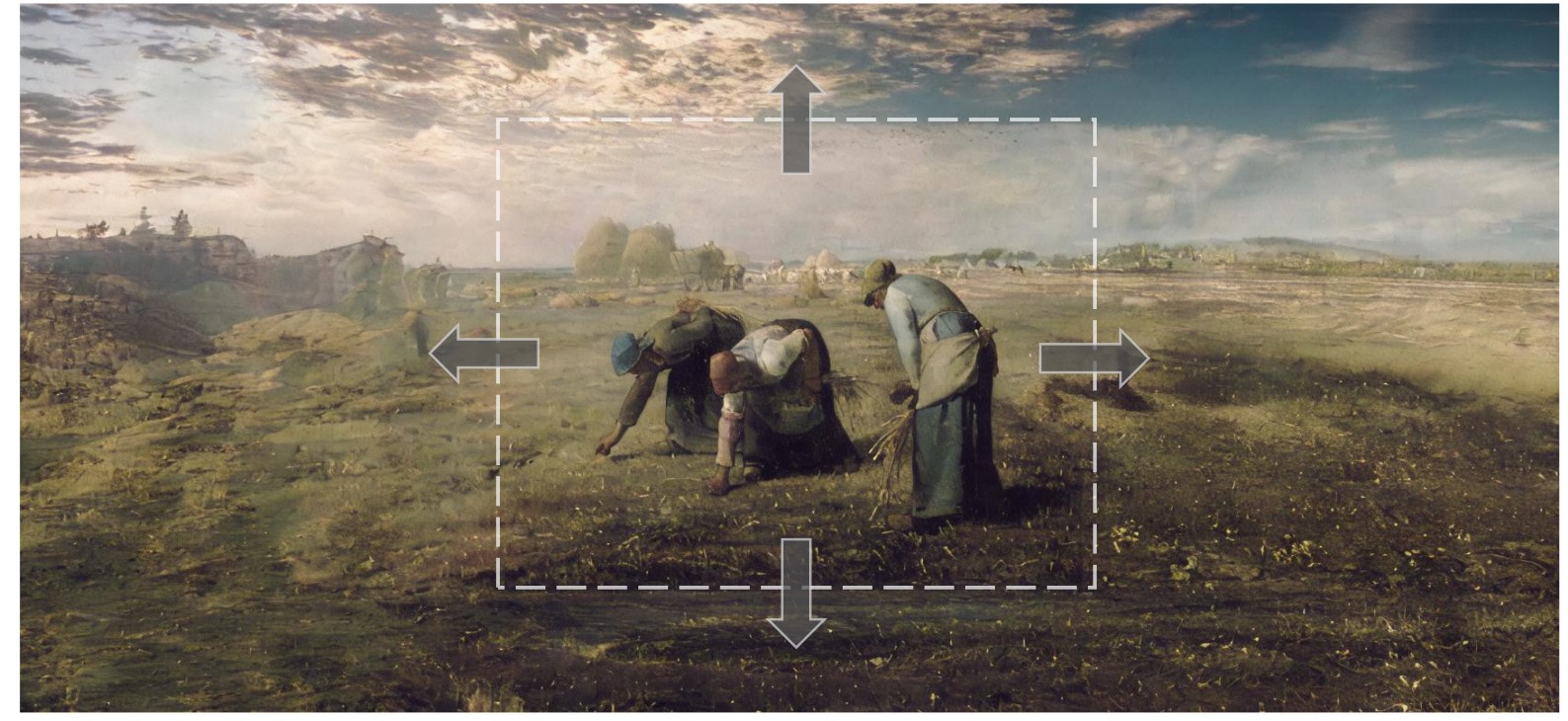

Figure 9: Arbitrary Image Extension 1536×3328 samples.

Input

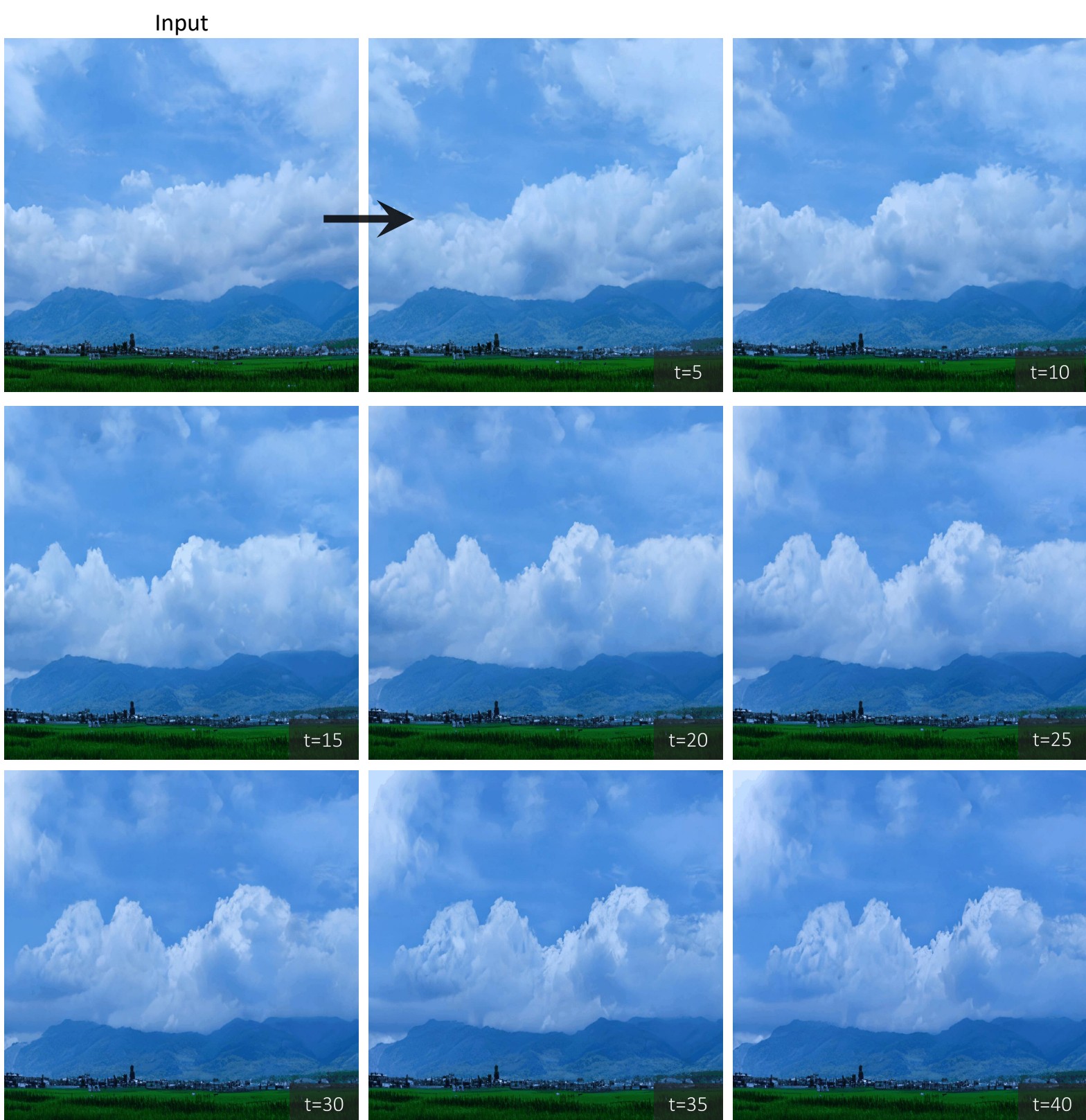

Figure 10: Image-to-Video 1024×1024×40 samples

Input

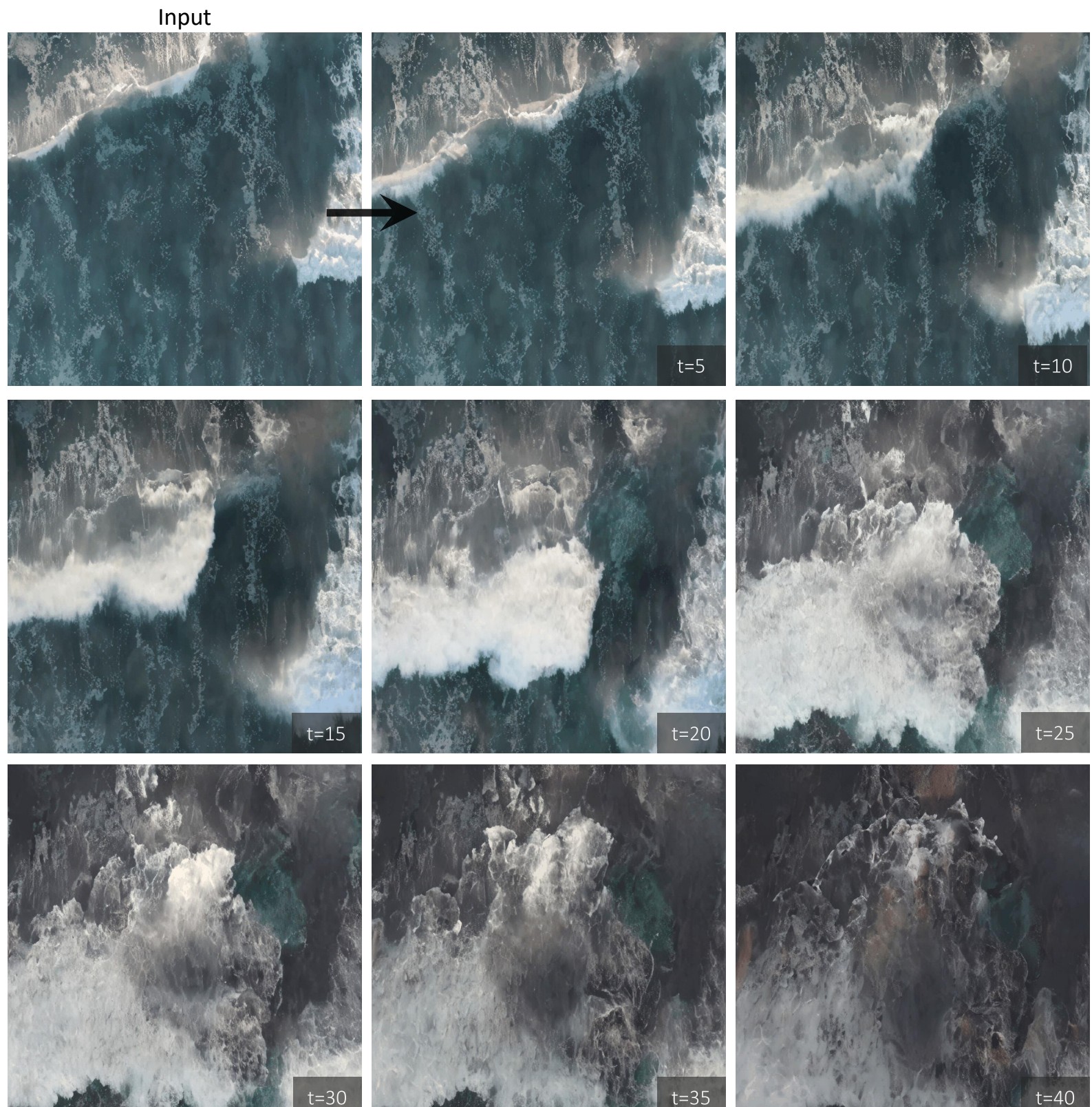

Figure 11: Image-to-Video 1024×1024×40 samples