# OpenReview forum: "NUWA-Infinity: Autoregressive over Autoregressive Generation for Infinite Visual Synthesis"
_NeurIPS.cc/2022/Conference — NeurIPS 2022 Accept_

### Official Review · Reviewer_msGb · 2022-06-26

**Rating:** 7
**Confidence:** 2
**Soundness:** 3 good
**Presentation:** 3 good
**Contribution:** 3 good

**Summary:**

The paper proposes an infinite visual synthesis approach using patch-level rendering and optimization. The main contribution includes (1) a "render-and-optimize" strategy; (2) arbitrary direction extension; (2) achieving good performance compared with baseline results.

**Questions:**

How is Fig. 1 generated? What is the input? It would be better if details can be included in the caption.

**Limitations:**

The limitations are mentioned in the supplementary materials. But it would be good if the authors mention them briefly in the main paper.

**Strengths And Weaknesses:**

Strengths:

+ The proposed "render-and-optimize" strategy is novel and effective.

+ The experimental results are significantly better than baseline results.

+ The ablation study is extensive and solid.

Weaknesses:

- Although 5 shows video frames individually, the video itself is not included in the supplementary material. This makes it hard to see the temporal smoothness, which is usually critical for video quality evaluation.

- The algorithm is claimed to be fast. However, there's no evaluation about speed, and the factors affecting the speed.

---

> ### Author Response · Authors · 2022-08-02
> **Responses to Reviewer msGb**
>
> ***
> ### Raw video not included in supplementary material
> Thanks for the suggestion, we will put the raw video into the paper. We also provide videos linked by https://sites.google.com/view/videopredictionbypro to show the video prediction results of PRO.
>
> ***
> ### Speed of PRO
> The following table shows the inference speed on unconditional image generation. The speed that affects our framework is mainly rendering model.  PRO provides a speed-performance trade-off rendering model, AR rendering model has the best performance but sacrificed the generation speed，NAR rendering model has fast inference speed，but parallel generation causes visual quality degradation.
> Table1: Inference Speed on Text-to-Image task
>
> | Method | Inference time↓|  |  |
> | ---- | ---- | ---- | ---- |
> | | 1024X1024 | 4096X1024 | 8192X1024 |
> | Taming | 286.7 x	 | 1232.4 x | 2580.0 x |
> | MaskGIT | 0.48 x | 11.6 x	 | 27.2 x |
> | PRO(AR) | 93.1 x | 372.4 x | 744.8 x |
> | PRO(NAR) | 6.3 x | 25.2 x | 50.4 x |
> | PRO(PNAR) | 1.0 x	 | 4.0 x | 8.0 x |
>
> As image resolution increases, our patch rendering strategy based on context pool is faster than overlapping sliding windows strategy such Taming and MaskGIT.It is worth mentioning that PRO(AR) still significantly faster than Taming(AR), this is because our model renders image without overlapping, however taming can only get context through overlapping areas.
>
> ***
> ### How is Fig. 1 generated?
> Firstly, we build a special dataset, Riverside of Qingming Festival (RQF), based on the version of "Along the River During the Qingming Festival" drawn by Qiu Ying. Then we resize the whole image to 66270×2048 resolution and split it into several overlapped 2048×2048 patches instead of non-overlapping ones. We train our model with the < ∅, image > pairs on the dataset with 2048×2048 resolution, Finally, images with 38912×2048 resolution are generated unconditionally by PRO starting from a BOS token.
>
> Thanks for your suggestion, we will add these details into the caption of Fig.1 in our paper.
> ***
> ### Move limitation Discussion from Supplementary Material into Main paper
> Thanks for the suggestion, We will add brief limitations in the main paper in the next version.

---

> > ### Comment · Reviewer_msGb · 2022-08-09
> > **Reply**
> >
> > Thanks for your detailed response. The rebuttal addressed my concerns. I will keep my acceptance rating.

---

### Official Review · Reviewer_j1NF · 2022-07-06

**Rating:** 5
**Confidence:** 5
**Soundness:** 3 good
**Presentation:** 2 fair
**Contribution:** 2 fair

**Summary:**

The paper proposes a render-and-optimize framework that supports autoregressive models to synthesize in arbitrary directions. Some memory components are being proposed to keep track of longer-term dependencies.

**Questions:**

See above.

**Limitations:**

I think the main problem and comparisons are not well discussed and compared. It is impossible to discuss the limitations without resolving the previous issues.

**Strengths And Weaknesses:**

Strengths:
1. The concept of synthesize in arbitrary direction is the strongest point of the paper. But it is not very well-discussed in the paper.
2. The visual results are with decent qualities. But it is not clear what are the proposed components that contribute/improve the visual quality, and what are the challenges that prior methods (InfinityGAN[17] and ALIS[30]) fail to address.

Weaknesses:
1. Some statements are pretty strong, but lacking results backing up. For instances,
(a) In the abstract, "However, since they fail to model global dependencies between patches, the quality and consistency of the generation can be limited." Some methods, such as InfinityGAN, model the global dependencies among all pixels with a global latent variable. But there is no comparison or any results supporting the statement is correct. The paper is cited in the first paragraph of the introduction but omitted in the second paragraph.
(b) In conclusion, "This strategy allows us to generate images and videos of infinite size." This is not the task that the authors first propose or address, which causes a mistaken over-claim. The authors should instead consider discussing what has been improved with the proposed framework.

2. The paper emphasizes the method takes care of the global dependency, but the algorithm design, such as the nearby context pool and the rendering model, only attends to the local region within a limited number of steps. Could the authors clarify which part of the pipeline models the global dependencies? It should be highlighted and carefully ablated if it is a critical intuition and contribution.

3. The comparisons to important baselines are missing, InfinityGAN and ALIS[30]. Especially, both of the baselines compared in the submission (VQGAN[8] and MaskGIT[2]) were not designed for infinite visual synthesis. The authors should compare with more relevant baselines.

4. The paper proposes modules that keep track of memory states, but there are no experiments showing the memory mechanisms are working properly. Especially, consider methods such as ALIS[30] that works completely fine without memory. The authors should demonstrate the importance of memory, and perhaps show that memory states indeed affects the results in distant patches. For instance, compare the differences between with and without a distant patch, is it just some random perturbations, or is it indeed provide some long-range dependency information?

5. The paper features infinite visual synthesis, but all the provided samples are only a few steps of extension. Could the author provide some visual results after 1k steps? The intermediate results can be omitted to save space. It is important to show that the algorithm can still produce plausible results after a large number of steps.

6. The additional applications in image-to-video and image-outpainting are all presented without careful comparisons. These are not new applications that lack baselines. For video synthesis, some recent video synthesis frameworks (e.g., StyleGAN-V) can already synthesize infinite-length videos. The authors should compare with them in the video synthesis problem. And the image outpainting is also relevant to InfinityGAN.

7. Many citations are broken.

8. (Minor citation error) To the best of my knowledge, the first divide-and-conquer generation paper is COCO-GAN, but only its follow-up works are cited.

---

> ### Author Response · Authors · 2022-08-02
> **Responses to Reviewer  j1NF**
>
> ***
> ### Some statements are strong
> Thank you for your suggestion, we will add the corresponding description in the second paragraph, and adjust the statement of some conclusions to avoid misunderstanding.
>
> ***
> ### How global distribution is modeled?
> Theoretically, $p_{i}\sim P(p_{i}|c_{i},y) $ Where the generation of each patch $p_{i}$ is based on the context $c_{i}$ and an optional text prompt $y$. If each context $c_{i}$ is generated on all previous patches $p_{<i}$, $c_{i} \sim P(c_{i}|p_{<i})$ where makes the generation condition $c_{i}$ of each patch $p_{i}$ a global hidden state, which is obviously the global distribution of the modeling. However, We assume that the context of each patch can only reach $k$ distances, where $k \geq 1$. The following equation gives an how each context is calculated:
> $c_{1} \sim P(c_{1}|p_{[max(0,1-k),1)}) = P(c_{1}|p_{0})$
> $c_{2} \sim P(c_{2}|p_{[max(0,2-k),1)}) = P(c_{2}|p_{1}) ~~ if ~ k = 1 ~~~ or~~~ P(c_{2}|p_{0}p_{1}) ~~ if ~ k \geq 2 $
> $ ...$
> $ c_{1+k} \sim P(c_{1+k}|p_{[max(0,1+k-k),1+k)}) = P(c_{1+k}|p_{[1,1+k)}) $
> $ ... $
> $ c_{1+2k}  \sim P(c_{1+2k}|p_{[max(0,1+2k-k),1+2k)}) = P(c_{1+2k}|p_{[1+k,1+2k)}) $
> $ ...$
> $ c_{i} \sim P(c_{i}|p_{[max(0,i-k),i)})$
> Since each context $c_{i}$ contains the information of the previous $k$ patches, and each patch $p_{i}$ is generated by its context $c_{i}$ at the moment. Thus the hidden state of each patch $p_{i}$ contains the information of the previous $k$ patches, and hidden states of patches can be transferred between contexts:
> $c_{1} \rightarrow ... \rightarrow c_{1+k} \rightarrow ... \rightarrow c_{1+2k} \rightarrow ... \rightarrow c_{i}$
> Therefore, as long as $k \geq 1$, this chain will make the first patch information transfer to the last patch.
>
>
> Experimentally, In our Tab.4 (d) (Page 5 of our paper), We also discuss the impact of information transfer on visual quality. If we do not use information transfer in our proposed PRO model, the performance significantly drops. This gives experiment evidence that the information transfer is crucial, as it models the global probabilistic instead of independent patches. Moreover, the global consistency of the Fig.1 (Page 1 of our paper) with 38912×2048 resolution also shows the benefits of information transfer and implicit global probabilistic distribution modeling.
>
>
> ***
> ### Comparisons between InfinityGAN/ALIS and PRO.
> Thanks for this suggestion. We compare InfinityGAN and ALIS in the following table. Since both InfinityGAN and ALIS do not support text-to-image task, we instead conduct experiments on unconditional image generation on LHQ1024 dataset. We find that PRO outperforms ALIS and InfinityGAN with a significant Block-FID score of 10.17 (1024x1024 resolution) and Block-FID(x4) score of 14.73 (4096x1024 resolution).
> Table1: Unconditional Image Generation on LHQ1024:
>
> | Method | Block-FID↓| Block-FID(x4)↓|
> | ---- | ---- | ---- |
> | ALIS | 19.65 | 20.54 |
> | InfinityGAN  | 23.67 | 27.18 |
> | PRO | 10.17 | 14.73 |
>
>
> ***
> ### The influence of the memory is not discussed
> We discussed it in  Fig.7 (b) (c) (Page 9 of our paper) , which demonstrate the impact of spatial memory and temporal memory on model performance, respectively.
>
> ***
> ### Extremely long results are not provided
> We provided it in Fig.1 (Page 1 of our paper), which shows a result with a resolution of 38912×2048, which consists of 1216 steps(patches) of 256×256 resolution. We also provide more samples linked by https://sites.google.com/view/samples-using-long-steps-by-pr
>
>
> ***
> ### Comparisons between InfinityGAN/StyleGAN-V and PRO
> Thank you for your suggestion. We compare the InfinityGAN and StyleGAN-V on the LHQC(1024×1024 resolution, train 85K, test 5K) and LHQ-V(1024×1024 resolution, train 38K, test 2K) datasets, respectively, here are the quantitative comparison results. For Image Outpating, Our proposed PRO get best performance in four directions. For Video Prediction, PRO also get a significant FVD socre of 62.57.
>
> Table2: Image Outpainting on LHQC1024:
>
> | Method | Block-FID↓|  |  |  |
> | ---- | ---- | ---- | ---- | ---- |
> |  | Right Extend	 | Left Extend | Down Extend | Up Extend |
> |InfinityGAN w/o text| 17.93 | 19.76 | 24.62 | 23.59 |
> | PRO w/o text  | 6.43 | 6.71 | 11.47 | 8.03 |
> | PRO w/ text	 | 6.45 | 6.72 | 9.84 | 7.43 |
>
>
>
> Table3: Video Prediction on LHQ-V:
>
> | Method | FVD↓|
> | ---- | ---- |
> | StyleGAN-V | 143.76 |
> |PRO| 62.57 |
>
>
> ***
> ### Broken Citations/Minor Citation Errors
> Thanks for the reminder, we will fix the citations and cite COCO-GAN as the first divide-and-conquer generation model.

---

> > ### Comment · Reviewer_j1NF · 2022-08-08
> > **Responses to rebuttal**
> >
> > ### How global distribution is modeled?
> >
> > My main concern is the term "global" is not entirely correct. The importance of distant contexts becomes less important as the sequence gets longer. Yes, theoretically, in the ideal case, with infinite capacity, the network can model the $c_1 \rightarrow \dots \rightarrow c_k$ relation. But in reality, take the 38912×2048 image generation for example, will the last-step synthesis results be different if the context of the first step is removed? This type of sequence modeling still keeps the information local. The global information should be something equally applied everywhere without any consideration of contextual distance, for instance, the artistic style or the tone of colors.
> >
> > ### Comparisons between "InfinityGAN/ALIS and PRO" & "InfinityGAN/StyleGAN-V and PRO"
> >
> > Please include the results in the final paper, and add some corresponding discussion. My major concern is that the positioning of this paper is not very clear with respect to the existing literature. The additional results can largely help readers understand the relation between this paper and the previous literature.
> >
> > ### Extremely long results are not provided
> >
> > Please add those results in the supplementary, and properly reference Figure 1 in the main paper. Clearly describing the setup (patch sizes and how many steps) in the caption of Figure 1 will be helpful as well.
> >
> > ### Notes
> >
> > I have no further questions. I will update the rating to 5, as the additional experimental helps address the major concerns (the position of this paper w.r.t. existing literature). But some parts of the writing may need further polishing.

---

### Official Review · Reviewer_q8JV · 2022-07-11

**Rating:** 6
**Confidence:** 4
**Soundness:** 3 good
**Presentation:** 3 good
**Contribution:** 3 good

**Summary:**

This work presents an infinite image and video synthesis method based on a render and optimize strategy. The proposed autoregressive model iteratively predicts next patch according to contexts in order to take the dependency between patches into account. The generated results look less repetitive and more coherent than those by existing methods. And the property of infinite generation step is a good progress towards generating images in ultra-resolution compared to fixed size generation through a big generation network like GANs.

**Questions:**

See comments above.

**Limitations:**

No. I'm wondering if the proposed method mainly works well on generating scene or landscape. Generating complex object shape should be still challenging.

**Strengths And Weaknesses:**

+ Modeling the global distribution of a large visual data so that each generated patch has taken the context into consideration.

+ Impressive results in ultra-resolution which is rarely seen in previous work

+ The proposed method is demonstrated both in spatial dimension of images and temporal dimension of videos.

Overall, it is a nice work that brings the extrapolation or outpainting task to a new level. I have following concerns:

(1) It looks the proposed pipeline share similar idea with some early autoregressive image generation models like PixelRNN or PixelCNN which are generating each pixel iteratively based on the previously generated pixels, though this work is operating on the patch level. What are the main differences compared with these related works?

(2) Is the proposed work mainly for unconditional generation, based on an optional text prompt? Could it be used for conditional generation, like image extrapolation to extend its original field of view?

(3) To what extent could the long-range dependency between patches be learned? Or how far is the relation of two patches modeled? I understand it might be not across all image regions but it is better to show and analysis on this so that it could help control the generation steps or image resolution for the best quality.

---

> ### Author Response · Authors · 2022-08-02
> **Responses to Reviewer q8JV**
>
> ***
> ### Difference between PixelCNN and PRO
> Although both PixelCNN and PRO are auto-regressive models, there are three main differences:
> * Different auto-regressive grains. PixelCNN is a pixel-by-pixel autoregressive model, while PRO is a patch-by-patch autoregressive model.
> * Different auto-regressive directions. PixelCNN is only one-directional thus it can only out-painting images on a single direction. In contrast, PRO supports all different directions and it out-paint an image on arbitrary directions.
> * Different optimization strategy. PixelCNN optimizes after all the pixels are generated, while PRO uses a render-and-optimize strategy. As shown in Tab.4(e) (Page 9 of our paper), we also show that the latter strategy significantly helps convergence and performance.
>
> ***
> ### Does PRO supports image extrapolation?
> Yes. In this paper, we call “image extrapolation” by “image extension”, and we provide multiple image extension results in Fig.7, Fig.8 and Fig.9 (Page 8, 9 and 10 of Supplementary Material).
>
> ***
> ###  what extent could the long-range dependency between patches be learned?
> Theoretically, $p_{i}\sim P(p_{i}|c_{i},y) $ Where the generation of each patch $p_{i}$ is based on the context $c_{i}$ and an optional text prompt $y$. If each context $c_{i}$ is generated on all previous patches $p_{<i}$, $c_{i} \sim P(c_{i}|p_{<i})$ where makes the generation condition $c_{i}$ of each patch $p_{i}$ a global hidden state, which is obviously the global distribution of the modeling. However, We assume that the context of each patch can only reach $k$ distances, where $k \geq 1$. The following equation gives an how each context is calculated:
> $c_{1} \sim P(c_{1}|p_{[max(0,1-k),1)}) = P(c_{1}|p_{0})$
> $c_{2} \sim P(c_{2}|p_{[max(0,2-k),1)}) = P(c_{2}|p_{1}) ~~ if ~ k = 1 ~~~ or~~~ P(c_{2}|p_{0}p_{1}) ~~ if ~ k \geq 2 $
> $ ...$
> $ c_{1+k} \sim P(c_{1+k}|p_{[max(0,1+k-k),1+k)}) = P(c_{1+k}|p_{[1,1+k)}) $
> $ ... $
> $ c_{1+2k}  \sim P(c_{1+2k}|p_{[max(0,1+2k-k),1+2k)}) = P(c_{1+2k}|p_{[1+k,1+2k)}) $
> $ ...$
> $ c_{i} \sim P(c_{i}|p_{[max(0,i-k),i)})$
> Since each context $c_{i}$ contains the information of the previous $k$ patches, and each patch $p_{i}$ is generated by its context $c_{i}$ at the moment. Thus the hidden state of each patch $p_{i}$ contains the information of the previous $k$ patches, and hidden states of patches can be transferred between contexts:
> $c_{1} \rightarrow ... \rightarrow c_{1+k} \rightarrow ... \rightarrow c_{1+2k} \rightarrow ... \rightarrow c_{i}$
> Therefore, as long as $k \geq 1$, this chain will make the first patch information transfer to the last patch.
>
>
> Experimentally, In our Tab.4 (d) (Page 5 of our paper), We also discuss the impact of information transfer on visual quality. If we do not use information transfer in our proposed PRO model, the performance significantly drops. This gives experiment evidence that the information transfer is crucial, as it models the global probabilistic instead of independent patches. Moreover, the global consistency of the Fig.1 (Page 1 of our paper) with 38912×2048 resolution also shows the benefits of information transfer and implicit global probabilistic distribution modeling.
>
> ***
> ### Could PRO generate non-landscape images?
> Yes.
>  In Fig.1 (Page 1 of our paper), “The RiverSide of Qingming Festival” generates many characters in various poses and costumes and various houses and buildings.
>
> We also provide a link https://sites.google.com/view/moresamplesbypro with many other kinds of samples  generated by PRO, such as LSUN-Church, a dataset of abstract design images and even Peppa Pig Videos.

---

> > ### Comment · Reviewer_q8JV · 2022-08-10
> > **Reply**
> >
> > Thanks for the response from authors. Better to include the above discussion and analysis in the revised version. I will keep my acceptance rating. It is a good work.

---

### Official Review · Reviewer_o5hC · 2022-07-14

**Rating:** 5
**Confidence:** 4
**Soundness:** 3 good
**Presentation:** 3 good
**Contribution:** 2 fair

**Summary:**

This paper focuses on visual synthesis, especially for high-resolution cases. To model global dependencies between patches,
this work proposes a patch-level render-and-optimize strategy, i.e., once a patch is predicted, its hidden states are saved as contexts for the next round of prediction. In addition, an arbitrary direction relative position embedding method is proposed, which shows promising performance on free-direction visual extension. The experimental results show the good performance of the proposed method on visual synthesis.

**Questions:**

Besides the questions in the weaknesses, here are some other questions.
1. As claimed, the proposed method brings two advantages (i) The autoregressive
rendering process with information transfer between contexts provides an implicit
global probabilistic distribution modeling. Why could provide an implicit global probabilistic distribution modeling? Any evidence could be provided?
2. The visual synthesis could only generate similar cases with the training data or could generate some out-of-distribution cases? What is the generalization capability? For example, does it generate any natural scenes?


**Limitations:**

Yes, the authors adequately the limitations of their work.

**Strengths And Weaknesses:**

**Strengths**

+Clear Motivation
The motivation of this paper is clear. The story flow is smooth.

+Interesting idea
Although the render-and-optimize strategy is intuitive, the idea is interesting and reasonable.

+Goog performance
From the shown results, the performance of the proposed method is promising.

+Sufficient experiments
This paper shows sufficient experimental results in the main paper and supplementary material. It would be good if more baselines could be included for comparison.


**Weakness**

-Difference from previous works
Although the paper claimed the differences between the proposed method and several previous methods, the specific differences are not clear.
For example, this work also uses the VQGAN as the Taming. Are there any differences between VQGANs? This work is also like MAE. Is it possible for MAE to generate such infinite visual synthesis? More explanations about the uniqueness are expected.

-Technique questions
The proposed render-and-optimize strategy first generates the first patch, then eventually generates other patches taking previous patches as conditions, like a Markov chain. Thus, the first patch is significant for the final result. It seems like a basis. What the results would be like if the first patch prediction is not much good? Such analysis is necessary and also reflects the robustness.

---

> ### Author Response · Authors · 2022-08-02
> **Responses to Reviewer o5hC**
>
> ***
> ### Difference between Taming and PRO
> Both PRO and Taming uses VQGAN as backbone to convert raw pixels into discrete tokens. However, PRO considers the dependency between different patches of a large image or video and uses a render-and-optimize strategy to train them jointly. While Taming assumes different patches are i.i.d. and train them separately.
>
> ***
> ### Difference between MAE and PRO
> Although MAE can reconstruct images with masks, but it usually generates blurry results(As shown in Fig.2 and Fig.3 of MAE paper). Also, MAE can only generate images with limited size of 224×224 or 448×448. While PRO can generate extremely large images with even 38912 × 2048 resolution such as Fig.1 (Page 1 of our paper).
>
> ***
> ### What the results would be like if the first patch prediction is not much good?
> Yes, you are right. AR models typically suffer from error-propagation. Although PRO is a patch-by-patch autoregressive model globally, PRO alleviates error-propagation issue by “render-and-optimize” strategy, where each loss of the patch is backward and optimized. We give  some example linked by https://sites.google.com/view/samplesfromfirstpatch, where the first patch prediction is not good, but the overall results still looks reasonable.
>
> ***
> ### Why PRO provides implicit global probabilistic distribution modeling?
> Theoretically, $p_{i}\sim P(p_{i}|c_{i},y) $ Where the generation of each patch $p_{i}$ is based on the context $c_{i}$ and an optional text prompt $y$. If each context $c_{i}$ is generated on all previous patches $p_{<i}$, $c_{i} \sim P(c_{i}|p_{<i})$ where makes the generation condition $c_{i}$ of each patch $p_{i}$ a global hidden state, which is obviously the global distribution of the modeling. However, We assume that the context of each patch can only reach $k$ distances, where $k \geq 1$. The following equation gives an how each context is calculated:
> $c_{1} \sim P(c_{1}|p_{[max(0,1-k),1)}) = P(c_{1}|p_{0})$
> $c_{2} \sim P(c_{2}|p_{[max(0,2-k),1)}) = P(c_{2}|p_{1}) ~~ if ~ k = 1 ~~~ or~~~ P(c_{2}|p_{0}p_{1}) ~~ if ~ k \geq 2 $
> $ ...$
> $ c_{1+k} \sim P(c_{1+k}|p_{[max(0,1+k-k),1+k)}) = P(c_{1+k}|p_{[1,1+k)}) $
> $ ... $
> $ c_{1+2k}  \sim P(c_{1+2k}|p_{[max(0,1+2k-k),1+2k)}) = P(c_{1+2k}|p_{[1+k,1+2k)}) $
> $ ...$
> $ c_{i} \sim P(c_{i}|p_{[max(0,i-k),i)})$
> Since each context $c_{i}$ contains the information of the previous $k$ patches, and each patch $p_{i}$ is generated by its context $c_{i}$ at the moment. Thus the hidden state of each patch $p_{i}$ contains the information of the previous $k$ patches, and hidden states of patches can be transferred between contexts:
> $c_{1} \rightarrow ... \rightarrow c_{1+k} \rightarrow ... \rightarrow c_{1+2k} \rightarrow ... \rightarrow c_{i}$
> Therefore, as long as $k \geq 1$, this chain will make the first patch information transfer to the last patch.
>
>
> Experimentally, In our Tab.4 (d) (Page 5 of our paper), We also discuss the impact of information transfer on visual quality. If we do not use information transfer in our proposed PRO model, the performance significantly drops. This gives experiment evidence that the information transfer is crucial, as it models the global probabilistic instead of independent patches. Moreover, the global consistency of the Fig.1 (Page 1 of our paper) with 38912×2048 resolution also shows the benefits of information transfer and implicit global probabilistic distribution modeling.
>
> ***
> ### Could PRO generate out-of-distribution cases?
> Yes. As shown in Fig.9 (Page 10 of Supplementary Material), PRO successfully out-paints an out-of-distribution case “The Starry Night”. Note that PRO only trains in real-world landscapes instead of any paintings. This shows a strong zero-shot capability our model.

---

### Comment · Area_Chair_BvQx · 2022-08-09
**reminder for discussion**

Dear reviewers,

Thank you all for providing valuable comments. The authors have provided detailed responses to your comments. Has the response addressed your major concerns?

I would appreciate it a lot if you could reply to the authors’ responses soon as the deadline is approaching (Tues, Aug 9).

Best,

ACs

---

### Meta-Review · Area_Chair_BvQx · 2022-08-28

**Recommendation:** Accept
**Confidence:** Certain

**Metareview:**

This paper proposes a patch-level autoregressive model for infinite visual synthesis based on two extensions: (1) transfer information across patches via context vectors and (2) timely optimization for each patch. Note that hierarchical autoregressive models have been explored in previous works such as VQ-VAE2 (https://arxiv.org/pdf/1906.00446.pdf). But the idea of context pool and arbitrary direction modeling seems interesting. The paper has received consistently positive reviews. Reviewers found the idea intuitive and the results visually compelling. The rebuttal further addressed the concerns such as the missing comparisons (w/ InfinityGAN and ALIS), running speed, and clarifications w.r.t existing works. The AC agreed with the reviewers’ consensus and recommended accepting the paper.

**Award:**

No

---

### Decision · Program_Chairs · 2022-09-14

Accept